# Visually Descriptive Language Model for Vector Graphics Reasoning

## Abstract

Despite significant advancements, current large multimodal models (LMMs) struggle to bridge the gap between low-level visual perception—focusing on shapes, sizes and layouts—and high-level language reasoning involving semantics, events and logic. This limitation becomes evident in tasks requiring precise visual perception, such as comparing geometric properties or solving visual algorithmic reasoning problems. To study this failure mode, we focus on an important visual domain: vector graphics—images composed purely of 2D objects and shapes, which are prevalent in various LMM-based agent tasks in web, visual design, and OS environments. We identify two key research questions: how can we enable precise visual perception, and how can we facilitate high-level reasoning based on such low-level perceptions? To accurately capture low-level visual details, we utilize Scalable Vector Graphics (SVG) for precise encoding of visual scenes. However, SVGs are not readily interpretable by LLMs or LMMs in a zero-shot manner. To address this challenge, we propose the **Visually Descriptive Language Model (VDLM)**, which introduces an intermediate textual representation called **Primal Visual Description (PVD)**. PVD translates SVGs into a text-based abstraction comprising primitive attributes (e.g., shape, position, measurement) along with their corresponding values. PVD can be learned with task-agnostic synthesized data and represents visual primitives that are universal across various vector graphics. This abstraction is more structured, allowing for direct interpretation by foundation models for zero-shot generalization to different reasoning tasks. Without any human-annotated data, empirical results demonstrate that VDLM leads to significant improvements in state-of-the-art LMMs, such as GPT-4o, across various low-level multimodal perception and reasoning tasks on vector graphics. Additionally, we provide extensive analyses of VDLM's performance, showing that our framework offers improved interpretability due to its disentangled perception and reasoning processes. Finally, we demonstrate the promise of this representation by showing a positive correlation between the quality of the PVD perception and the end-task performance.

## 1 Introduction

In recent years, large multimodal models (LMMs) (OpenAI, 2023b; Anil et al., 2023; Liu et al., 2023b; Chen et al., 2023b; Bai et al., 2023) have achieved impressive performance across a broad spectrum of general vision-language benchmarks (Goyal et al., 2017; Fu et al., 2023; Liu et al., 2023d; Yu et al., 2023; Li et al., 2023a). However, these monolithic LMMs still struggle with seemingly simple tasks that require precise perception of low-level visual details. In particular, we empirically observe that LMMs frequently exhibit this failure mode in vector graphics, which are images composed purely of 2D objects and shapes, devoid of any camera viewpoint. For example, a state-of-the-art LMM like GPT-4o (OpenAI, 2024) can still fail 43% of the time when comparing the lengths of two line segments, and 54% of the time when solving a simple 2×2 maze. LMMs' ability to understand vector graphics is largely underexplored compared to natural images but is essential for growing downstream applications, such as LMM-based agents in web, visual design, and OS environments (Zhou et al., 2023b; Liu et al., 2024; Xie et al., 2024; Rawles et al., 2024; Zheng et al., 2024; Lù et al., 2024). To address this challenge, we identify two main research questions. First,

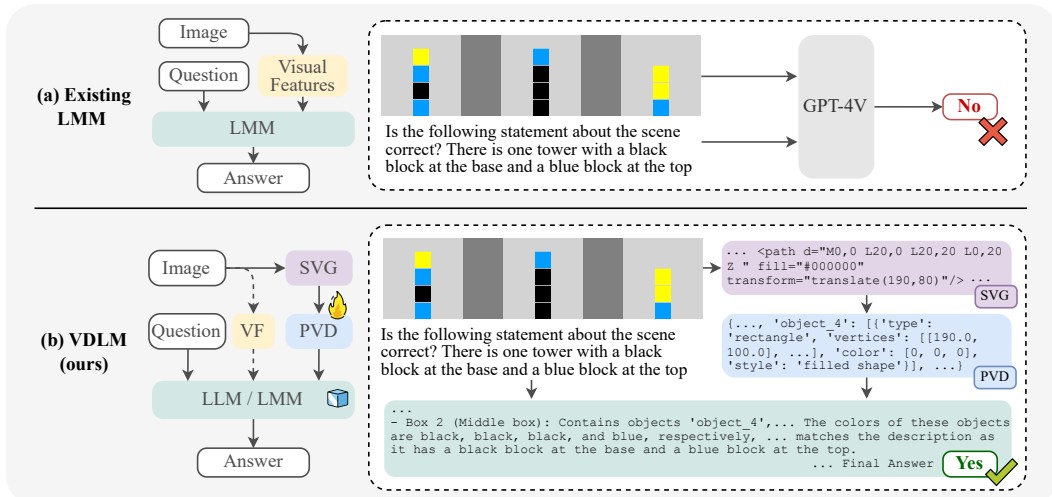

Figure 1: Existing monolithic LMMs rely solely on pretrained vision encoders, such as CLIP (Radford et al., 2021), for perception, which often fail to accurately capture low-level visual details in vector graphics. In contrast, VDLM enables precise visual reasoning by first encoding the input image into SVG format and then learning an intermediate symbolic representation, Primal Visual Description (PVD), that bridges low-level SVG perception with high-level language reasoning.

how can we enable precise visual perception in LMMs? Second, how can we effectively leverage low-level visual perception for vision-language reasoning?

For our initial question, we explore vectorizing a rasterized image using the Scalable Vector Graphics (SVG) representation, which describes a scene with paths (e.g., polygons and splines) and their corresponding measurements and positions. SVG representations, by nature, are unbiased towards high-level semantics and can capture low-level visual details in text. The vectorization process can be faithfully accomplished with an off-the-shelf, rule-based raster-to-vector algorithm. However, despite being text-based, SVG is insufficient for language reasoning. Our preliminary experiments (§A) demonstrate that existing large language models (LLMs) are unable to interpret machine-generated SVG codes in zero-shot settings. Moreover, finetuning a model to reason about raw SVG codes can be inefficient and infeasible without corresponding task-specific annotations.

To address the challenge posed in our latter question, we propose training a language model to align the extracted SVG paths to an intermediate symbolic representation, which can directly be leveraged by foundation models such as LLMs or LMMs for low-level visual reasoning. We introduce **Primal Visual Description (PVD)**, which bridges the low-level SVG codes and the high-level language space for reasoning about vector graphics. Specifically, we train an LLM-based (Jiang et al., 2023) SVG-to-PVD model, which transforms the raw SVG paths into a set of primitive attributes (e.g., shape, position) with corresponding predicted values (e.g., rectangle, pixel coordinates of the vertices). See Figure 1 in the blue box for an example. Notably, the PVD representation contains primitive attributes that are universal across vector graphics, and thus can be learned with procedurally generated ⟨SVG, PVD⟩ pairs without task-specific annotations. Since PVD is more structured and closer to natural language, it allows for direct interpretation by pretrained foundation models.

Comprising SVG-based image perception and primitive-level abstractions, we present our method, the **Visually Descriptive Language Model (VDLM)**. VDLM has three components: a rule-based visual encoder that converts images to SVG to capture precise visual features, a learned language model that translates SVG to PVD, and an inference-only LLM or LMM reasoner that conducts zero-shot reasoning about downstream tasks with the PVD representation. For VDLM with LMM reasoners, we keep the original visual features of the input image and add the PVD representation seamlessly into the text prompt as additional visual descriptions. An overview of VDLM is provided in Figure 1.

Experimental results demonstrate that VDLM, using only PVD perception and a text-only LLM as the reasoner, can already achieve strong zero-shot performance in various visual reasoning tasks, out-

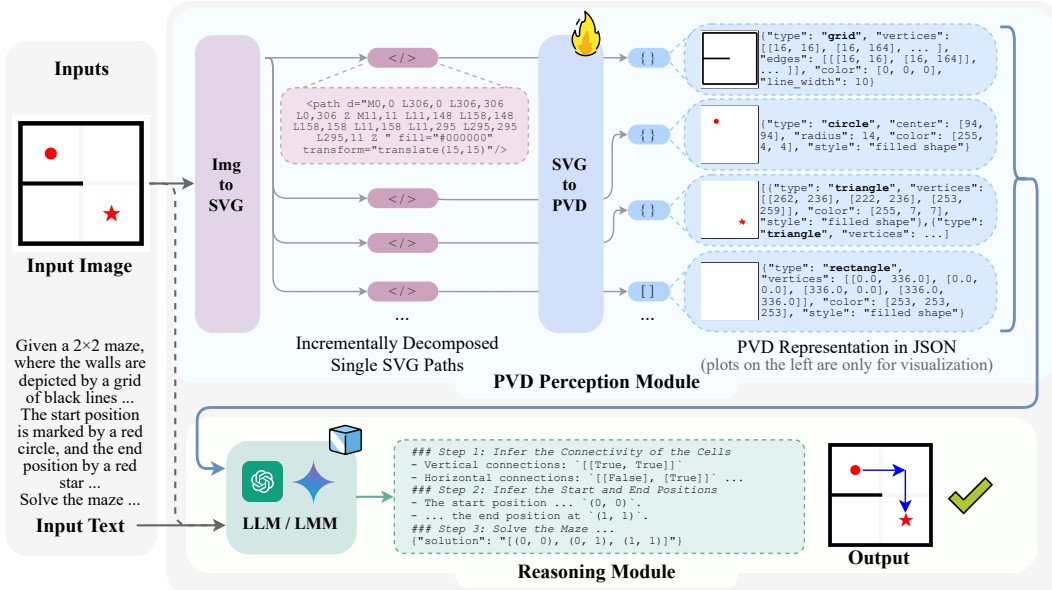

Figure 2: An example of VDLM during inference. First, VDLM extracts individual SVG paths from the input image and then transforms them into PVD descriptions using a newly learned language model. These PVD perception results, along with the input text queries and optionally the original input image, are subsequently fed into an LLM or LMM for reasoning. It is worth noting that although a "star" (★) is not explicitly part of the PVD primitive ontology (see Figure 3), the SVG-to-PVD model can approximate the "star" by composing two triangles (★). A strong off-the-shelf reasoner, such as GPT-4 (OpenAI, 2023a), can accurately deduce that this composition corresponds to the "star," which is the target end position of the maze. For the complete response, refer to Figure 14.

performing LLaVA-v1.5 (Liu et al., 2023a), G-LLaVA (Gao et al., 2023), GPT-4V (OpenAI, 2023b), and Visual Programming approaches such as ViperGPT (Surís et al., 2023). Furthermore, equipping VDLM with a strong LMM reasoner, such as GPT-4V (OpenAI, 2023b) or GPT-4o (OpenAI, 2024), brings significant improvement to vector graphics reasoning.

Importantly, VDLM also enhances interpretability through its disentangled perception and reasoning processes. We conduct an in-depth analysis of the impact of perception quality on the final task performance, revealing that more accurate PVD perception leads to improved overall performance. This underscores the promise of our disentangled framework, where the improvement of the perception module can directly lead to improvement of the entire system. Notably, our PVD representation is trained with only synthesized data and has limited coverage of concepts; we hope this work can inspire future work for building more general visually descriptive representations.

To summarize, the key contributions of our work are threefold: First, we identify a critical failure mode of LMMs when reasoning about tasks that require precise, low-level perception in vector graphics. Second, we introduce VDLM, a visual reasoning framework that operates with intermediate text-based visual descriptions—SVG representations and learned Primal Visual Description, which can be directly integrated into existing LLMs and LMMs. Finally, we show that VDLM can bring significant improvements to complex low-level reasoning about vector graphics with pretrained foundation models; our analysis also provides insights into the perception and reasoning steps of VDLM.

## 2 VDLM FRAMEWORK

We present the VDLM framework, which comprises three components. First, a rule-based perception module transforms images into SVG format, accurately capturing low-level visual details (§ 2.1). Second, a trained language model aligns SVGs with intermediate visual descriptions by mapping SVG paths to primitive shapes (§ 2.2). Third, an inference-only LLM or LMM reasons about the downstream tasks with the text-based perception results (§ 2.3). See Figure 2 for an overview.

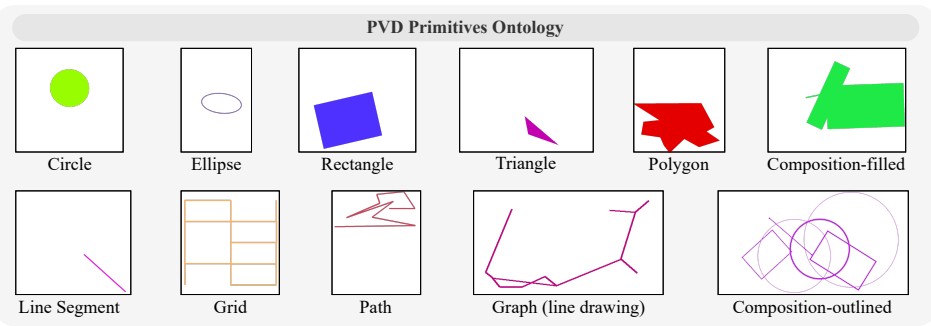

Figure 3: Ontology of the primitives in Primal Visual Description (PVD).

## 2.1 ENCODING IMAGES INTO SVG WITH RULE-BASED ALGORITHMS

Prior work (Krojer et al., 2022; Tong et al., 2024) has demonstrated that, although CLIP-based (Radford et al., 2021) vision encoders are effective at capturing high-level visual semantics, they can fall short in preserving fine-grained visual details. As an alternative, we propose extracting an SVG representation that more accurately captures the detailed measurements. Unlike raster graphics, such as JPEG or PNG images, which represent images through a grid of pixels, SVG describes shapes, lines, and colors using mathematical expressions and paths with precise coordinates. We posit that these distinctions allow SVG representations to more faithfully describe visual scenes in vector graphics.

To empirically verify this, we conduct a suite of preliminary experiments (§ A) investigating the potential of using SVG for representing visual inputs. We find that on vector graphics tasks, fine-tuning the LLM backbone, Vicuna (Chiang et al., 2023), of an LLaVA-1.5 (Liu et al., 2023a), with SVG representations consistently outperforms fine-tuning the entire LLaVA model with CLIP-based features. Importantly, we can leverage a rule-based raster-to-SVG parsing algorithm (VTracer) for converting any image into SVG without learning. This enables us to obtain an unbiased initial description of the visual input. However, we observe two key challenges (§ A.3) when working with raw SVG representation. First, off-the-shelf LLMs, e.g., GPT-4 (OpenAI, 2023a), have a limited zero-shot reasoning ability on SVG representation. Even with fine-tuning, training an LLM to directly understand raw SVG code can still be challenging. Second, fine-tuning on task-specific ⟨SVG, question, answer⟩ pairs limits generalization to unseen tasks and domains. We discuss our approach of extracting intermediate representations below.

## 2.2 LEARNING ALIGNMENT OF SVG TO PVD WITH LANGUAGE MODELS

**Primal Visual Description (PVD).** We propose Primal Visual Description, a higher-level abstraction that transforms low-level SVG paths to more structured primitives required for reasoning. PVD is a text-based visual description that consists of a set of primitive geometry objects, e.g., circles and line segments. Each PVD element contains the primitives' attributes (e.g., color, shape, position, size) with corresponding predicted values (e.g., blue, circle, pixel coordinates of the center, length of the radius). An example of the PVD representation is as follows (See Figure 13 for full definitions):

```
{"type": "circle", "center": [252, 315], "radius": 202,
"color": [175, 155, 98], "style": "filled shape"}
```

Notably, the PVD is a higher level of abstraction that can be extracted from SVG, from which we can directly leverage the strong reasoning abilities of an off-the-shelf LLM or LMM to generalize across various downstream tasks. Moreover, the PVD is general enough to serve as a unified visual description across different types of vector graphics, as most complex concepts can be composed of multiple primitive shapes. For example, a "cross" can be composed of two "rectangles."

As shown in Figure 3, the ontology of the Primal Visual Description contains 9 primitive shape types that can be composed to cover diverse vector graphics in the wild. The primitive shapes include circles, ellipses, rectangles, triangles, polygons, line segments, grids, paths, and graphs. A path

in PVD is defined as a non-intersecting polyline. Graphs and grids are defined as a set of vertices connected by a set of edges.

**Learning alignment with a language model.** We then train a language model to generate PVD outputs from SVG inputs. The input is a single SVG path depicting a visual concept, and the output is the predicted one or more primitives in the defined PVD ontology. During inference, given an arbitrary raster image, we first convert it into a raw SVG file, which may contain a large number of SVG paths, including unimportant noise and speckles. To denoise the raw SVG file and extract salient shapes, we propose an incremental decomposition algorithm. Specifically, we incrementally include SVG paths while checking the difference between the partially rendered image of currently chosen paths and the fully rendered image of the original raw SVG file. We compute the summation of the absolute pixel-by-pixel difference between the two images and set an empirical threshold. If the difference after adding a new path is below this threshold, i.e., if the added path does not bring much additional visual information to the scene, we will skip that path. For the ordering of the path selection, we follow the default ordering from VTracer that heuristically places the paths with a larger area at the front. The paths that come afterward will be stacked on top of previous paths during rendering. Upon obtaining the decomposed single SVG paths, we first generate their PVD representation individually. We then aggregate the individual PVD predictions into a holistic perception of the entire image using this JSON template: `["object_0":    <PVD output for path 0>, "object_1":    <PVD output for path 1>, ...]`.

Importantly, since PVD is task-agnostic, the data for training the SVG-to-PVD model can be procedurally generated without human annotation. We develop a data generator leveraging PIL.ImageDraw[*] and VTracer, which creates a large-scale ⟨SVG, PVD⟩ paired dataset containing randomly generated primitives. In some real-world tasks, such as geometry problems, multiple primitive shapes with the same color can overlap. When converted to SVG, these shapes tend to be parsed into one merged SVG path. To enable the SVG-to-PVD model to learn to decode individual primitives from such compositional concepts, we additionally generate data instances with randomly overlapped shapes. The target PVD representation, in this context, is a list of primitive PVD JSON objects. We ensure that each generated image contains only one unicolor object, single or composed, so that the converted SVG contains a single SVG path. This facilitates a language model in effectively learning the alignment between SVG and PVD.

To improve the robustness to unseen inference images, we randomize the image sizes, the positions and rotations of the shapes, as well as the styles of the shapes (filled or outlined). We additionally use two data augmentation methods, Gaussian Blur and Pixel Noise, to add variance to the training SVG paths. Our final dataset contains 160K ⟨SVG, PVD⟩ pairs. More details can be found in Appendix C.

We fine-tune a pretrained Mistral-7b (Jiang et al., 2023)[†] model on the synthesized PVD 160K dataset to perform SVG-to-PVD generation. We conduct full-parameter fine-tuning for 3 epochs with a learning rate of 1e-5. The training objective is a standard Language Modeling loss on the generated PVD tokens as follows:

$$\mathcal{L} = -\frac{1}{N} \sum_{i=1}^{N} \log P(\mathbf{d}_i | \mathbf{s}, \mathbf{d}_{0:i-1}) \tag{1}$$

where $\mathbf{s}$ and $\mathbf{d}$ refer to the input SVG tokens and the generated PVD tokens respectively. We use the Megatron-LLM (Cano et al., 2023) library for efficient LLM fine-tuning and the entire training process can be done in 16 hours on 4 NVIDIA A100-40GB GPUs.

## 2.3 REASONING ABOUT PRIMAL VISUAL DESCRIPTION WITH LLMs AND LMMs

Our visual perception modules generate a fully text-based visual description from the input vector graphics image. For each downstream task, we input the perception result into the prompt along with the task-specific instructions, and then feed it into an inference-only LLM or LMM reasoner.

We explore two variants of VDLM, namely **VDLM-txt** and **VDLM-mm**, depending on the type of reasoner applied. VDLM-txt leverages a text-only LLM as the reasoner and solely uses Primal Visual Description to represent the visual information, whereas VDLM-mm leverages an multimodal

---

[*]https://pillow.readthedocs.io/en/stable/reference/ImageDraw.html
[†]https://huggingface.co/mistralai/Mistral-7B-v0.1

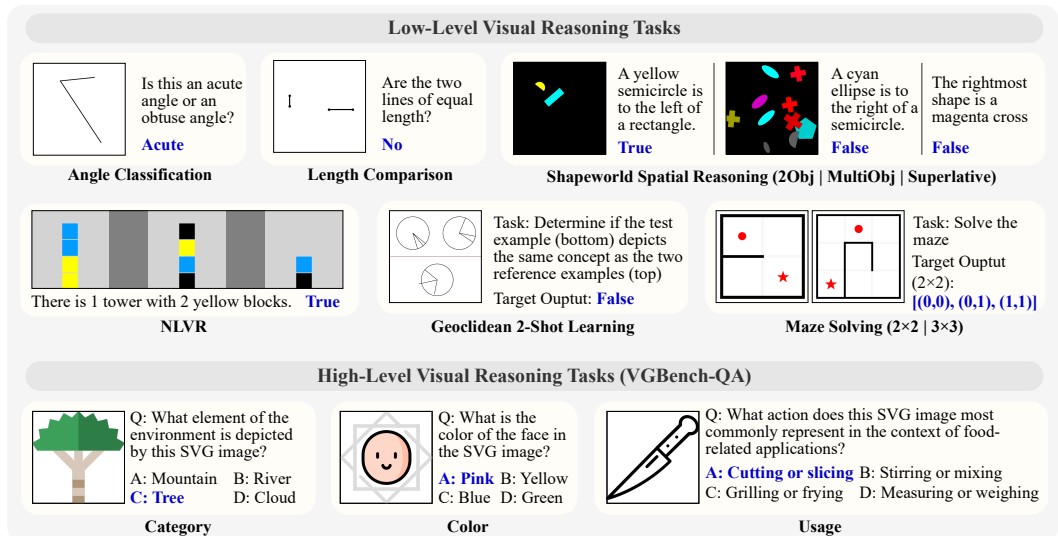

Figure 4: Our full evaluation benchmark with a focus on **low-level visual reasoning** about vector graphics (detailed in 3.1). We additionally include high-level reasoning tasks with rendered SVG images from VGBench-QA (Zou et al., 2024). All tasks are evaluated in a zero-shot setting.

LMM as the reasoner, which can additionally take the original image as visual input. A detailed execution trace of the VDLM functions is illustrated in Figure 2. We observe that a strong reasoner, such as GPT-4 (OpenAI, 2023a), without any fine-tuning, can effectively perform various types of task-specific reasoning based on the PVD representation. This includes identifying higher-level concepts, computing measurements, examining spatial relations, and performing multi-step reasoning. The reasoning procedure is also more explainable and transparent compared to the output of existing monolithic LMMs.

## 3 EXPERIMENTS

### 3.1 TASKS

**Low-level visual reasoning tasks.** Evaluating LMMs in tasks that require precise visual perception about vector graphics is a highly underexplored research area and has limited existing resources. To this end, we construct a new evaluation benchmark that comprises 9 tasks which cover important aspects of low-level visual perception and reasoning, including measurements, spatial relations, counting, logical reasoning, and complex reasoning problems such as maze solving. The description of each task is as follows: (1) **Angle Classification**: Identify whether an angle is acute or obtuse. (2) **Length Comparison**: Determine whether two line segments are of equal length. (3-4) **Shapeworld Spatial Reasoning**: The Shapeworld (Kuhnle & Copestake, 2017) dataset on spatial relations with images containing exactly two objects or multiple objects. (5) **Shapeworld Superlative**: The Shapeworld dataset on superlative statements. (6) **NLVR**: The Natural Language for Visual Reasoning dataset (Suhr et al., 2017) which contains diverse counting, spatial reasoning, and logical reasoning queries. (7) **Geoclidean 2-shot Learning**: A repurposed Geoclidean (Hsu et al., 2022) dataset requiring the model to understand a compositional geometric concept with only two reference examples. (8-9) **Maze Solving**: Solve a 2×2 or 3×3 maze, given the starting and ending positions. Among these tasks, Angle Classification, Length Comparison, and Maze Solving are newly created from scratch (See Appendix F for more details).

**High-level visual reasoning tasks.** Although the focus of this work is on low-level visual reasoning, we additionally include a set of high-level tasks to investigate the impact of VDLM on knowledge reasoning tasks. These tasks rarely require precise perception of the locations and measurements of the primitives. We leverage VGBench (Zou et al., 2024), a benchmark originally proposed for evaluating LLMs in understanding and generating vector graphics codes. In this work, we evaluate LMMs and VDLM-mm for question-answering based on the rasterized VGBench SVG images.

Figure 4 shows simplified input and output examples for each task. Full prompts can be found in Appendix E. To reduce the cost of evaluating proprietary models, we randomly sample a subset of

| Low-level Visual Reasoning on Vector Graphics | | | | | | | | | | |
|---|---|---|---|---|---|---|---|---|---|---|
| | Tools | AC | LC | SW-S 2Obj | SW-S mObj | SW Sup | NLVR | Geo | Maze 2×2 | Maze 3×3 | All |
| Monolithic Large Multimodal Models | | | | | | | | | | | |
| Llava-1.5-7b | - | 0.53 | 0.49 | 0.48 | 0.55 | 0.35 | 0.53 | 0.50 | 0.00 | 0.00 | 0.381 |
| Llava-1.5-13b | - | 0.53 | 0.51 | 0.51 | 0.47 | 0.61 | 0.48 | 0.50 | 0.00 | 0.00 | 0.401 |
| Gllava-7b | - | 0.59 | 0.50 | 0.43 | 0.54 | 0.43 | 0.49 | 0.58 | 0.00 | 0.00 | 0.396 |
| GPT-4V | - | 0.58 | 0.64 | 0.77 | 0.60 | 0.61 | 0.63 | 0.64 | 0.28 | 0.02 | 0.530 |
| GPT-4o | - | 0.63 | 0.57 | **0.97** | **0.82** | **0.92** | 0.81 | **0.71** | 0.46 | 0.08 | 0.663 |
| Visual Programming with LLM (text-only) reasoner | | | | | | | | | | | |
| ViperGPT (w/ GPT-4) | CI | 0.11 | 0.67 | 0.61 | 0.47 | 0.53 | 0.43 | 0.02 | 0.03 | 0.00 | 0.319 |
| VDLM with LLM (text-only) reasoners | | | | | | | | | | | |
| **VDLM-txt** (w/ GPT-4) | - | 0.89 | 0.95 | 0.78 | 0.63 | 0.80 | 0.68 | 0.63 | 0.40 | 0.19 | 0.661 |
| **VDLM-txt** (w/ GPT-4) | CI | 0.73 | 0.95 | 0.89 | 0.68 | 0.72 | 0.72 | 0.64 | 0.40 | 0.26 | 0.666 |
| VDLM with LMM (multimodal) reasoners | | | | | | | | | | | |
| **VDLM-mm** (w/ GPT-4V) | - | 0.55 | 0.94 | 0.84 | 0.62 | 0.72 | 0.71 | 0.69 | 0.60 | 0.20 | 0.652 |
| **VDLM-mm** (w/ GPT-4o) | - | **0.90** | **0.95** | 0.91 | **0.82** | 0.82 | **0.86** | 0.71 | **0.61** | **0.34** | **0.769** |

Table 1: Zero-shot accuracy on low-level visual reasoning tasks. Task abbreviations: AC (Angle Classification), LC (Length Comparison), SW-S-2Obj/mObj (Shapeworld Spatial Reasoning with two objects or multiple objects), SW-Sup (Shapeworld Superlative), Geo (Geoclidean 2-shot Learning). "CI" refers to Code Interpreter. Notably, VDLM-txt, with text-only reasoning, already outperforms strong LMMs such as GPT-4V. Compared to GPT-4V, GPT-4o shows enhanced capability particularly in spatial reasoning, but still struggles with simple primitives such as angles and lines. VDLM-mm brings consistent overall improvements to GPT-4V and GPT-4o by simply incorporating PVD as additional textual prompt. The remaining negative impacts arise from limitations in PVD perception, as well as the reasoner's capability. Detailed analysis is presented in §3.3 and §4.

100 instances for each task. We consider a zero-shot evaluation setting for all tasks. Note that the SVG-to-PVD model in VDLM is trained purely on synthesized task-agnostic data and has not seen any downstream tasks.

## 3.2 MODELS

We compare our work with strong base­lines, including both state-of-the-art mono­lithic large multimodal models (LMMs), i.e., LLaVA-v1.5 (Liu et al., 2023a), GLLaVA Gao et al. (2023), GPT-4V (OpenAI, 2023a)[‡], GPT-4o (OpenAI, 2024)[§], as well as visual pro­gramming agents, e.g., ViperGPT (Surís et al., 2023). ViperGPT employs an LLM to gen­erate code, which can call external vision models, such as GLIP (Li et al., 2022) and BLIP2 (Li et al., 2023b), to process the im­age and generate the final output. Given that ViperGPT-style models successfully separate

| High-level Visual Reasoning on Vector Graphics | | | |
|---|---|---|---|
| | VGBench-QA | | |
| | Category | Color | Usage | All |
| Llava-v1.5-7b | 0.26 | 0.32 | 0.27 | 0.283 |
| Llava-v1.5-13b | 0.32 | 0.43 | 0.39 | 0.380 |
| Gllava-7b | 0.16 | 0.33 | 0.21 | 0.233 |
| GPT-4o | 0.58 | 0.84 | **0.76** | 0.726 |
| **VDLM-mm** (w/ GPT-4o) | **0.62** | **0.86** | 0.75 | **0.743** |

Table 2: Zero-shot accuracy on high-level visual reasoning tasks. We show that VDLM-mm preserves the LMM's capability on semantic-centric reasoning that does not require precise low-level perception.

perception from reasoning, we seek to investigate whether the existing perception tools adequately recognize low-level primitives in vector graphics. For VDLM, we explore two variants, namely VDLM-txt with GPT-4 (text-only) [¶], and VDLM-mm with GPT-4V and GPT-4o. We also experiment with applying weaker LMM reasoners, such as LLaVA, to VDLM-mm. We find that interpreting PVD requires a certain level of text reasoning capability, and the benefits only emerge with strong LMMs, as shown in Figure 5. To obtain more insights in comparing with ViperGPT, we further investigate augmenting VDLM-txt with a Code Interpreter (CI). We employ the GPT-4 Assistant [‖] for our experiments, designating the code interpreter as the sole tool available. We use the same set of prompts for both VDLM-txt and VDLM-mm. See details about prompt design in Appendix E.

[‡]GPT-4V model version: gpt-4-1106-vision-preview.

[§]GPT-4o model version: gpt-4o-2024-05-13.

[¶]GPT-4 (text-only) model version: gpt-4-0125-preview.

[‖]https://platform.openai.com/docs/assistants/overview/agents

### 3.3 RESULTS

Table 1 shows the zero-shot accuracy for the evaluation tasks. We outline the key findings as follows:

**VDLM-txt, even without access to the original image, outperforms strong LMMs**, highlighting the efficacy of the intermediate PVD representation for precise low-level perception and reasoning. We also observe that strong text-only models can make well-reasoned assumptions to creatively interpret the text-based perception results or filter out unimportant information. For instance, as illustrated in Figure 2, it correctly infers the compositional object with two triangles as a "star". See Figure 14 for the complete response.

**VDLM-mm significantly improves LMMs on low-level reasoning tasks, while preserving their capabilities in high-level reasoning.** Table 1 shows that, without any task-specific fine-tuning, strong LMM reasoners can effectively incorporate the additional information provided by the PVD representation alongside the image input. Figure 5 further demonstrates that this benefit only emerges when the LMM has a certain level of text-reasoning ability and persists in state-of-the-art LMMs. For high-level reasoning tasks (Table 2), the improvement is more subtle, as the tasks focus on the semantics of the vector graphics, such as "what can this be used for?", which rarely require precise location or measurements of visual elements.

**QA performance on complex math problems does not necessarily reflect a faithful understanding of low-level visual concepts.** We observe that G-LLaVA (Gao et al., 2023), a model demonstrating strong performance on geometric

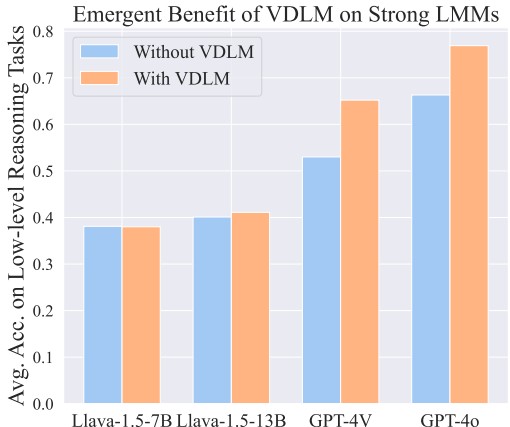

Figure 5: The direct improvements brought by VDLM to LMMs emerge when the LMM possesses sufficient text reasoning capabilities. These improvements are consistent with stronger LMMs, such as GPT-4o, which have enhanced spatial reasoning performance.

problems, such as MathVista (Lu et al., 2023), still struggles with understanding basic lines and angles, which are prerequisites for solving geometric math problems.

**Existing vision-language models, such as GLIP and BLIP2, are ineffective as low-level visual preceptors.** This is evidenced by the unsatisfactory performance of ViperGPT, even when equipped with a strong planner like GPT-4. On the other hand, we observe that augmenting the reasoning model in VDLM-txt with code interpreters can be particularly helpful for tasks requiring algorithmic reasoning, such as 3×3 maze solving.

**While our PVD provides a unified representation, there is potential to enhance its perceptual expressiveness.** In certain tasks, such as Shapeworld Spatial Reasoning, GPT-4o achieves better performance than VDLM-mm. The reason for this lies in the imperfect perception results from the SVG-to-PVD model. Since the SVG-to-PVD model is trained with purely synthetic data, it is not yet perfect when generalizing to diverse domains. We carefully analyze the remaining errors in §4.2, and demonstrate the impact of improving perception on end-task performance (§4.1). Future work is needed to develop a more general and expressive PVD representation.

## 4 ANALYSIS

### 4.1 PVD PERCEPTION QUALITY VS END-TASK PERFORMANCE

One advantage of a modular system is that enhancing an individual module can leads to improvements in the overall system. In this section, we explore whether a positive correlation exists between the quality of the intermediate perception representation and end-task performance. To investigate this, we first define metrics to reflect the quality of the Primal Visual Description (PVD) perception. Upon generating a PVD perception result, we render it back into a raster image using our procedural image generator. We then compute a similarity score between the reconstructed image and the original input image as a measure of the perception performance. For measuring the similarity, we consider various approaches, including both pixel-based and embedding-based metrics. We adopt

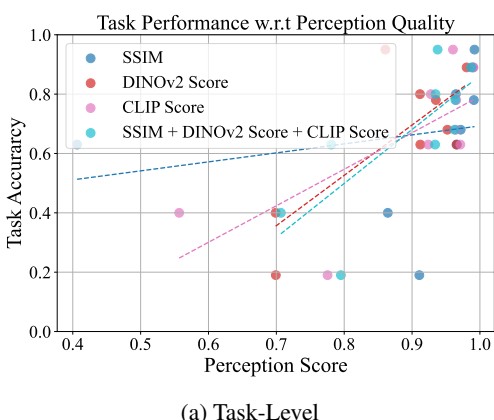

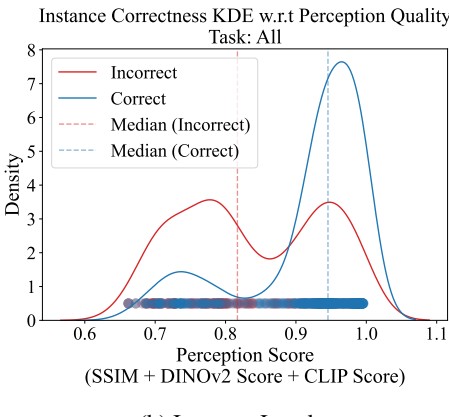

(a) Task-Level                                            (b) Instance-Level

Figure 6: Correlation between PVD perception quality and end-task performance. We observe a positive correlation between more precise perception and higher downstream accuracy.

the Structural Similarity (SSIM) Index (Wang et al., 2004) score to assess pixel-level similarity. Additionally, to account for semantic similarity, we adopt a CLIP-score (Radford et al., 2021) and a DINOv2-score (Oquab et al., 2023), which are calculated as the cosine similarity of the flattened CLIP and DINOv2 embeddings, respectively.

In Figure 6, we visualize the impact of the perception quality, on the 9 low-level reasoning tasks with VDLM-txt, at both the task and instance levels. In Figure 6a, each point denotes the accuracy of a task, with different colors representing different similarity metrics. The dashed lines depict linear regression results of the points, revealing a consistent positive correlation between perception quality and task accuracy across the metrics. Since the task-level accuracy may not be directly comparable across different tasks, we additionally perform an instance-level analysis using Kernel Density Estimation (KDE) on the correctness of all task instances with respect to their perception scores. As shown in Figure 6b, the "correct" distribution visibly skews to the area of higher perception scores, indicating that better perception tends to result in a correct final answer. This finding is promising, suggesting that enhancing the intermediate PVD representation, even with a fixed reasoning model, can effectively boost downstream task performance.

## 4.2 DEEP DIVE INTO FAILURE MODES: A TRANSPARENT ERROR ANALYSIS

The improved interpretability, resulting from PVD's disentangled perception and reasoning, allows us to conduct an in-depth analysis of the failure modes of VDLM. We examine the errors made by VDLM-txt in low-level reasoning tasks, where the PVD representation is the only perception accessible to the LLM reasoner. We find that both the perception step (SVG-to-PVD) and the reasoning step (PVD-to-answer) can contribute to errors. On tasks that require complex multistep reasoning, such as Maze Solving, reasoning errors become more prevalent; otherwise, perception errors most directly contribute to poor performance. Details and illustrative examples of these errors are provided in **Appendix B**, along with a distribution of perception and reasoning errors from human analyses. The prevalent error types for both perception and reasoning steps are summarized as follows.

Common perception errors include failures in faithfully perceiving novel shapes that are not covered by or cannot be composed within the PVD ontology; failures in capturing intentional constraints between primitives, such as a line exactly segmenting a circle, due to the random nature of the data generation on the positioning of objects; and failures in capturing very small objects, due to the heuristic thresholding in the incremental SVG decomposition algorithm. In Table 5, we show that the proposed augmentation during synthetic data generation improves PVD perception. However, we see that there is still a large room for improvement, by defining a more general visually descriptive representation, diversifying, and scaling up the data generation pipeline. We leave this to future work.

Common reasoning errors over the PVD perception include failures in discovering intentional constraints without being explicitly asked, such as automatically recognizing that a rhombus is not

the same concept as a general quadrilateral; failure in handling ambiguous instructions; and failure in complex multi-step reasoning tasks, such as solving mazes.

## 5    RELATED WORK

**Visual shortcomings in large multimodal models.**    While state-of-the-art LMMs achieve strong performance on existing multimodal benchmarks (Goyal et al., 2017; Fu et al., 2023; Liu et al., 2023b;d; Yu et al., 2023; Li et al., 2023a), which primarily focus on natural images, recent work (Lu et al., 2023; Yue et al., 2023; Huang et al., 2023; Zhou et al., 2023a; Hsu et al., 2022; Gao et al., 2023) has shown that they struggle with charts, geometric diagrams, and abstract scenes. This observation aligns with recent studies investigating visual shortcomings in LMMs. Tong et al. (2024) suggests that current LMMs struggle with visual details because the image-text contrastive pretraining of the CLIP visual backbone does not encourage the preservation of fine-grained visual features, such as orientation and quantity. To address this issue, recent studies have either leveraged the mixture-of-experts approach (Tong et al., 2024; Fan et al., 2024; Lu et al., 2024; Jain et al., 2023b), incorporating various types of vision encoders, such as SAM (Kirillov et al., 2023), DINOv2 (Oquab et al., 2023), or introduced auxiliary losses that emphasize local details during multimodal pretraining McKinzie et al. (2024); Bica et al. (2024); Varma et al. (2023). In this work, we propose a novel perspective for addressing this visual deficiency in vector graphics with an intermediate perception representation.

**Image vectorization and program synthesis.**    Generating vectorized or symbolic representations of visual concepts has been a topic of interest in both the NLP and computer vision communities. Recent work (Vinker et al., 2022; Lee et al., 2023; Ma et al., 2022; Rodriguez et al., 2023; Jain et al., 2023a; Tang et al., 2024; Xing et al., 2024; Hu et al., 2024) has investigated generating vector graphics codes from raster images or text prompts. In this work, we focus on the reverse problem of understanding and reasoning about vector graphics as visual inputs. We find that vector graphics reasoning serves as a challenging testbed to evaluate low-level visual reasoning abilities in large multimodal models (LMMs). Although Bubeck et al. (2023); Cai et al. (2023); Zou et al. (2024); Qiu et al. (2024) have shown initial promise in using large language models (LLMs) to understand the semantics of vector graphics codes, as shown in § A.3, they still struggle with understanding precise low-level details. Therefore, we propose the intermediate Primal Visual Description representation to further enhance low-level perception and reasoning, without sacrificing the performance of semantic understanding. This work is also heavily inspired by related work in neural-symbolic models (Ritchie et al., 2016; Wu et al., 2017; Yi et al., 2018; Mao et al., 2019; Hsu et al., 2024; Zhang et al., 2023; Trinh et al., 2024). This paradigm aims to de-render visual scenes into structured representations, retrieve programs from input text, and execute these programs on the image representations. Instead of defining task-specific symbolic programs, we extend the idea to learning a task-agnostic visual description that can be directly reasoned about by off-the-shelf foundation models for task generalization.

**Disentangling perception and reasoning in large multimodal models.**    Another closely related line of work has investigated disentangling visual perception and reasoning with visual programming (Gupta & Kembhavi, 2023; Surís et al., 2023; Ge et al., 2023; Wu & Xie, 2023) and tool-using (Wu et al., 2023; Liu et al., 2023c). These models leverage the code generation capabilities of LLMs to compose and employ a set of vision-language or vision-only models, such as object detection and image caption models, as subroutines for solving visual reasoning tasks. Despite promising performance on natural images, as shown in § 3, we find that these models are still limited by the existing vision-language models' inability to process low-level primitives effectively.

## 6    CONCLUSIONS AND FUTURE WORK

We present VDLM, a novel approach designed to address the limitations of large multimodal models in performing precise low-level perception and reasoning tasks in vector graphics. By leveraging SVG representations and introducing an intermediate symbolic abstraction, VDLM enables precise capture of low-level visual features as well as direct use of LLMs and LMMs for generalization. VDLM not only outperforms existing state-of-the-art LMMs such as GPT-4o but also enhances interpretability through its disentangled perception and reasoning process. The limitations of this work primarily stem from the capability of the SVG-to-PVD perception module. Although the PVD has already shown significant promise with a limited ontology and a fully synthesized training dataset, it is mainly designed for handling 2D vector graphics with basic primitives. Future directions include building a more general intermediate representation that has a broader coverage and can be extended from 2D vector graphics to 3D and natural images.

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

# SUPPLEMENTARY MATERIAL FOR VISUALLY DESCRIPTIVE LANGUAGE MODEL FOR VECTOR GRAPHICS REASONING

The appendix is organized as follows: In Appendix A, we present preliminary experiments comparing SVG and image-based representations. In Appendix B, we include details on error analyses, and in Appendix C, we describe Primal Visual Description details. Appendix D shows the full input and output from GPT-4 for the maze-solving example depicted in Figure 2. Task prompts and newly constructed downstream task datasets can be found in Appendices E and F, respectively. In Appendix G, we include detailed statistics for all of the datasets we used.

## A    PRELIMINARY EXPERIMENTS ON SVG REPRESENTATIONS

We introduce a suite of probing tasks to evaluate current LMMs' capabilities in performing tasks with vector graphics. The results show that even state-of-the-art LMMs, such as GPT-4V, struggle with tasks that require precise perception of low-level primitives, such as comparing the lengths of two lines. We then investigate where this deficiency originates and propose an alternative representation, Scalable Vector Graphics (SVG), for representing such precise low-level features. We find that, compared to image-based representations, SVG representations can be more efficient for visual reasoning on vector graphics. However, they are not without their own limitations, which we will elaborate on in § A.3.

### A.1    IMAGE AND SVG REPRESENTATIONS

In the probing tasks, we include both discriminative and generative tasks, each with varying levels of emphasis on low-level visual details. Illustrations of the input and output examples are available in Figure 7. We additionally include a non-vector-graphics task, Clevr QA, which consists of realistic 3D rendered scenes. This is to test the limits of SVG representations in encoding 3D objects within realistic images. Detailed statistics of these tasks can be found in Table 6.

For each task, we consider two evaluation settings: zero-shot and fine-tuning. We explore two types of representations for the input image: (1) direct use of the image pixels, encoding them into patch embeddings with an image encoder, e.g., CLIP (Radford et al., 2021); (2) conversion of the image into SVG code using a rule-based raster-to-SVG converter (VTracer).

For fine-tuning with the image input, we instruction-finetune Llava-v1.5-7b (including the LLM-backbone and the projection layer) using Lora (Hu et al., 2022) on the training set for one epoch. For fine-tuning with the SVG input, we only fine-tune the LLM backbone of Llava-v1.5, Vicuna (Chiang et al., 2023), using Lora for one epoch, with the input image's SVG code concatenated in the context. The results are shown in Table 3. Key observations include:

(1) The SOTA open-source LMM, Llava-v1.5, struggles to achieve non-trivial performance on most probing tasks even with dedicated fine-tuning. On tasks with binary choices, Llava tends to predict homogeneous answers, disregarding differences in the input image.

(2) The SOTA closed-source LMM, GPT-4V, excels on task Line or Angle, which focuses on querying the high-level semantics of the primitive concept ("what's in the image"). However, its performance significantly decreases on tasks requiring more precise low-level perception, e.g., Angle Classification and Length Comparison.

(3) Fine-tuning the LLM backbone, Vicuna, with SVG inputs consistently outperforms fine-tuning the entire Llava model with image inputs. This highlights the potential of using SVG as an alternative representation in vector graphics.

(4) We note that SVG may inherently be inefficient in representing rendered 3D scenes and realistic images due to factors like camera perspectives, lighting, and shadows. While our focus in this work is on vector graphics, we leave the extension to other domains for future exploration.

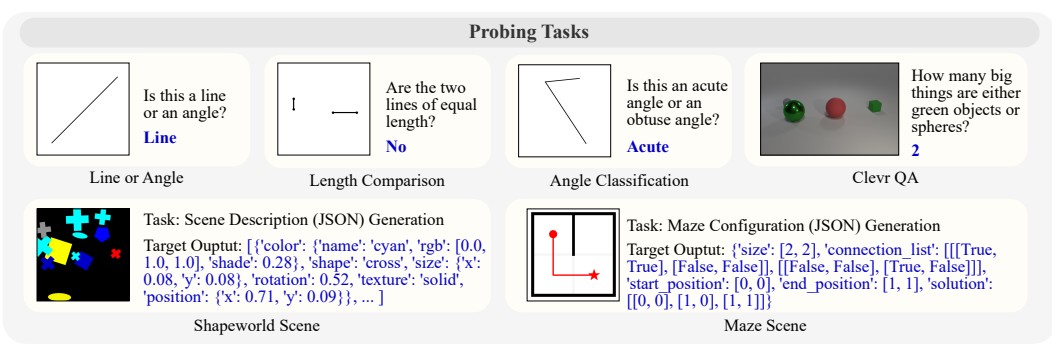

Figure 7: Illustration of the probing tasks. The four tasks at the top are question-answering tasks, while the two tasks at the bottom are scene-generation tasks. The goal of the scene-generation tasks is to generate the entire structured scene description following a predefined schema.

| | | Input Type | Line or Angle | Angle Classification | Length Comparison | Clevr QA |
|---|---|---|---|---|---|---|
| **Zero-Shot** | GPT-4V | Image | 1.00 | 0.58 | 0.57 | 0.57 |
| | GPT-4 | SVG | 0.45 | 0.47 | 0.60 | 0.36 |
| **Finetuned** | Llava-v1.5-7b | Image | 0.50 | 0.50 | 0.50 | 0.45 |
| | Vicuna | SVG | 0.93 | 0.70 | 0.99 | 0.54 |

| | | Input Type | Shapeworld Scene | | Maze Scene | |
| | | | shape (acc↑) | position (l2↓) | connectivity (acc↑) | start-pos (acc↑) | end-pos (acc↑) |
|---|---|---|---|---|---|---|---|
| **Zero-Shot** | GPT-4V | Image | 0.33 | 0.27 | 0.27 | 0.21 | 0.22 |
| **Finetuned** | Llava-v1.5-7b | Image | 0.04 | 0.67 | 0.26 | 0.03 | 0.03 |
| | Vicuna | SVG | 0.15 | 0.07 | 0.54 | 0.08 | 0.09 |

Table 3: Probing task results. We report the accuracy for the four question-answering tasks at the top. At the bottom, we use different metrics for different fields in the predicted scene description JSON. "acc" refers to accuracy (larger is better) while "l2" refers to the Euclidean distance between the predicted and ground truth [x, y] coordinates (lower is better). Scores with a blue background denote the better fine-tuned method compared to the SVG and Image representation. Scores with a red background denote tasks where fine-tuned methods cannot outperform zero-shot GPT-4V. Detailed analysis can be found in § A.1.

## A.2 LLAVA'S FAILURE MODE IN VISUAL REASONING WITH VECTOR GRAPHICS

We further investigate whether the difficulty in understanding low-level visual features of Llava models stems from (1) the visual backbone itself, i.e., CLIP, or (2) the bridge between the visual backbone and the LLM backbone. We include a set of **Linear Probing** experiments on three binary classification probing tasks, where we train a simple linear classifier based on the visual backbone features (before and after projection) of the Llava model. As shown in Figure 8:

(1) In tasks requiring more precise low-level perception, such as Angle Classification and Length Comparison, CLIP embeddings are inherently less effective at capturing relevant features. Furthermore, as shown in Figure 9, in some tasks, e.g., Length Comparison, linear regression even fails to achieve 90%+ training accuracy after 10 epochs of training, struggling to converge.

(2) When connected to an LLM using the projection layer, the visual features in Llava become less effective for low-level visual reasoning. Additionally, there is a significant gap between linear probing and instruction fine-tuning performance. These results suggest that even if the backbone does preserve useful features, the LLM cannot effectively leverage those visual tokens after projection.

We hypothesize that the failure mode likely stems from the multimodal pretraining and instruction-tuning paradigm, where the tasks are biased towards high-level semantics, such as image captioning (Lin et al., 2014; Sidorov et al., 2020) and natural-image-based VQA (Goyal et al., 2017; Krishna et al., 2017; Marino et al., 2019; Schwenk et al., 2022). The training mixtures (Liu et al., 2023b;a; Dai et al., 2023; Chen et al., 2023a) for current LMMs predominantly focus on high-level features

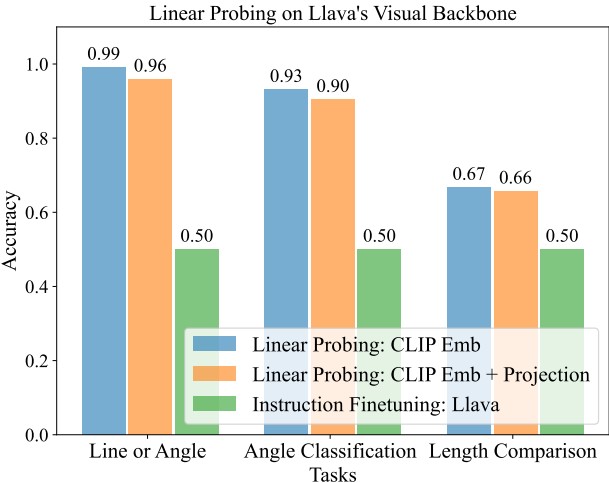

Figure 8: The average accuracy of linear probing, computed across ten epochs. Detailed training and testing scores for each epoch can be found in Figure 9. The results demonstrate that (1) CLIP embeddings are less effective for tasks requiring precise perception, such as Length Comparison, in comparison to tasks that emphasize on higher-level semantics, such as Line or Angle; (2) connecting to an LLM through the widely-used Llava-style architecture results in further diminished performance on tasks involving low-level visual details.

of images, providing little incentive for models to retain low-level visual details. For example, the caption of an image containing a 2D maze, such as the one shown in Figure 2, is likely to be "A 2×2 maze with black lines, a red circle and a star." and may not include detailed configurations of the mazes, such as the precise locations of the walls, the red circle, and the red star.

### A.3 REMAINING CHALLENGES OF USING SVG REPRESENTATIONS

Although we have demonstrated that SVG can serve as a promising alternative representation for reasoning about vector graphics, we identify several remaining challenges:

(1) Pretrained LLMs, including the most capable ones such as GPT-4 (OpenAI, 2023a), possess limited out-of-the-box understanding of SVG code. This limitation is evidenced by the low zero-shot performance of GPT-4 with SVG input (see row 2 in Table 3).

(2) Even after finetuning, the SVG-based LLM may still underperform zero-shot GPT-4V on certain tasks, particularly those involving complex scenes, such as Shapeworld Scene and Maze Scene. In these instances, the SVG code becomes excessively verbose. These findings suggest that learning a model to directly comprehend the raw SVG code of an entire image poses significant challenges.

(3) A fundamental challenge, irrespective of the chosen representation for visual input, is the lack of generalization capability to unseen tasks and various vector graphics image domains. If we rely on existing LMM training mixtures, even any image can be converted into SVG code, the tasks remain biased towards high-level semantics. In addition, it is infeasible to directly manually construct and annotate ⟨SVG, question, answer⟩ pairs covering diverse tasks with vector graphics.

These challenges motivated us to propose another layer of abstraction, the Primal Visual Description, aimed at bridging the gap between low-level perception and high-level language reasoning on downstream tasks.

## B ERROR ANALYSIS DETAILS

As introduced in § 2, the proposed VDLM consists of two stages focused on perception—namely, Image-to-SVG and SVG-to-PVD, and one stage focused on reasoning, i.e., PVD-to-final answer. We aim to investigate the errors in both the perception and reasoning modules.

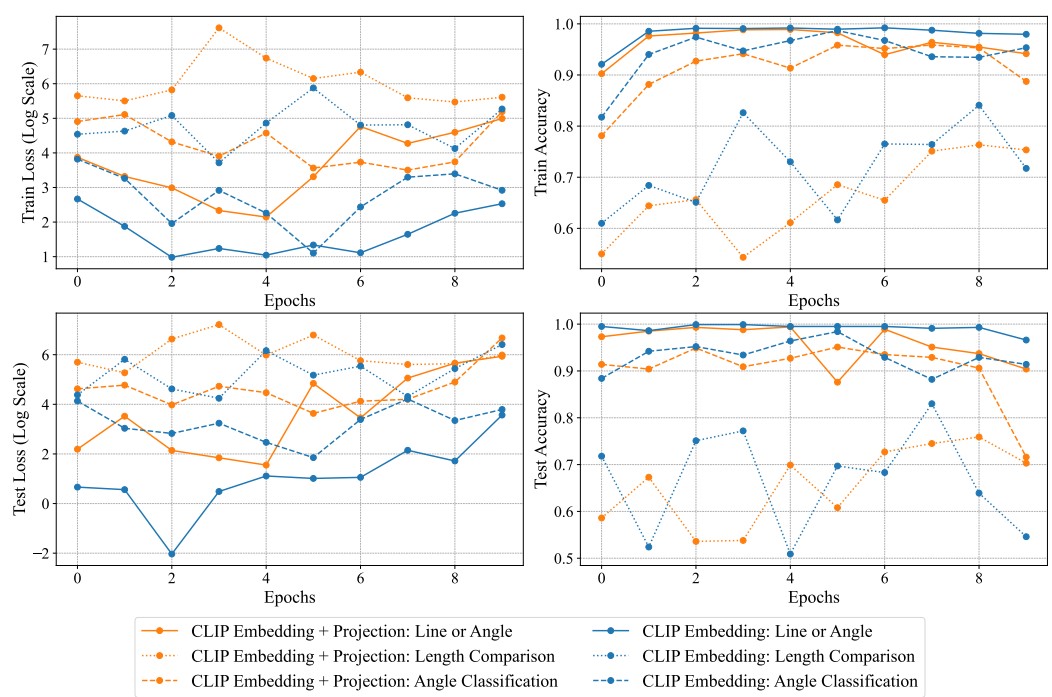

Figure 9: Linear probing training details: Different line styles represent different tasks, while different colors refer to different visual embeddings used for training the linear classifier. The training loss (top-left) shows that the projected embedding (orange lines) learns at a slower pace compared to the original CLIP embedding (blue lines). The training accuracy (top-right) reveals that for certain tasks, such as Length Comparison, the model continues to struggle with overfitting the training set even after 10 epochs.

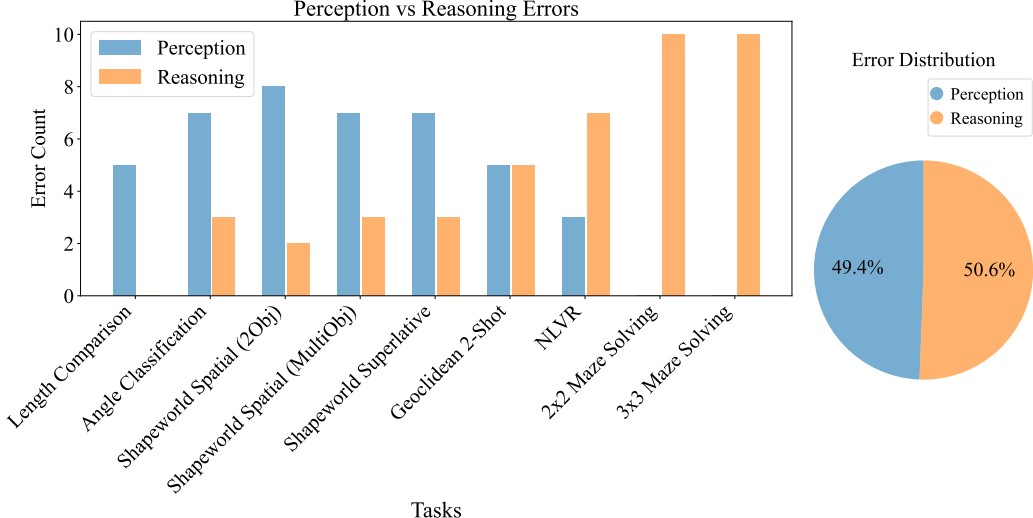

Figure 10: Error distribution by VDLM-txt between perception and reasoning on low-level vector graphics reasoning.

For each task, we manually examine 10 error cases and determine whether the error primarily stems from the perception stage or the reasoning stage. We task a human with reviewing the reconstructed image from the PVD representation and assessing the question of the task instance. If, for a human,

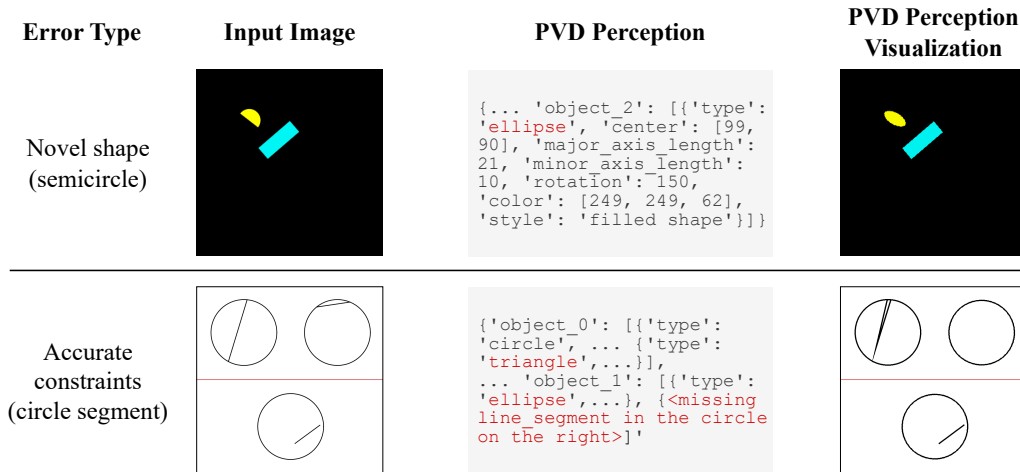

Figure 11: Perception error examples. The example at the top illustrates an error wherein the SVG-to-PVD model predicts a semicircle as an ellipse. The example at the bottom demonstrates that the SVG-to-PVD model struggles to decode overlapping primitives with accurate constraints, such as a segment of a circle.

the reconstructed image is still insufficient for solving the task, we classify this error as a perception error. Otherwise, it is categorized as a reasoning error. Figure 10 illustrates the distribution of errors between perception and reasoning stages. We further identify some typical categories of perception and reasoning errors as follows:

**Common perception errors.** (1) **Novel shapes not covered by the Primal Visual Description (PVD)**: For example, as visualized in Figure 11, the Shapeworld dataset includes a "semicircle" shape type which is not in the PVD ontology; we see that the learned SVG-to-PVD model tends to predict it as an ellipse. This perception error directly contributes to the inferior performance of VDLM-mm compared to GPT-4o on the Shapeworld tasks, as shown in Table 1.

(2) **Accurate constraints between primitives**: Although the PVD accommodates scenarios where multiple objects of the same color overlap, the attributes, e.g., position, of each object are decided independently and randomly. Thus, the SVG-to-PVD model often fails to capture intentional constraints between objects; for example, a line that perfectly segments a circle. These constraints are particularly emphasized in the Geoclidean 2-shot Learning task (Figure 11), where VDLM struggles to outperform GPT-4V and GPT-4o.

(3) **Very small objects**: During inference, the iterative decomposition process heuristically ignores SVG paths that only contribute only minor differences to the reconstructed image. This method effectively reduces noise from the rule-based image-to-SVG converter but may omit very small objects in some cases. Adjusting this threshold is necessary for specific scenarios.

**Common reasoning errors.** (1) **Discovering intentional constraints**: Without specific queries, the reasoning model can fail to identify intentional constraints. For example, differentiating a rhombus from a general quadrilateral, as shown in Figure 12.

(2) **Handling ambiguity**: Visual inputs sometimes provide useful inductive biases that can help the model better understand the task or make reasonable assumptions when the instructions are ambiguous. For instance, when presenting an angle in an image and asking whether it is an acute or obtuse angle, as in Figure 4, it is visually straightforward to assume that the angle is defined by the middle point as the vertex with rays extending outwards. However, without such visual cues, reasoning over pure symbolic representations makes it challenging to infer which angle the question refers to among the detected undirected edges. To mitigate ambiguity, adding more precise instructions for VDLM-txt is necessary in some tasks.

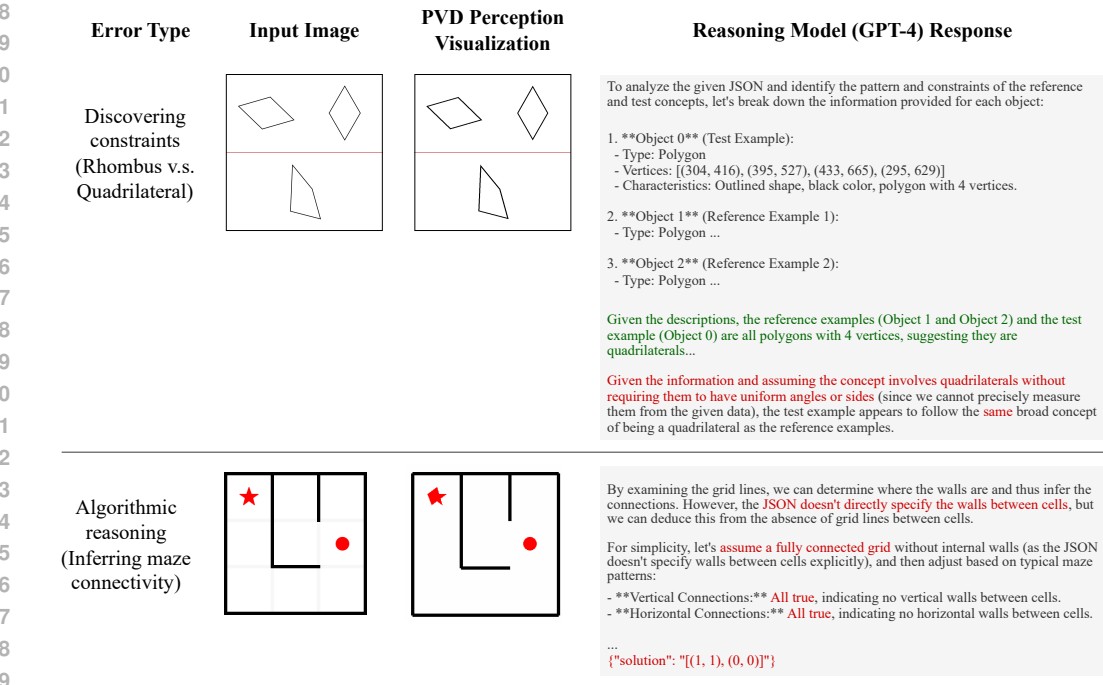

Figure 12: Reasoning error examples: The example at the top demonstrates that the reasoning model fails to uncover the deeper constraints within the perceived polygons. It is able to recognize that they are all quadrilaterals but unable to further discern that the reference concepts are rhombuses with four equal sides, while the test concept is not. The example at the bottom illustrates that the reasoning model struggles to infer connectivity based on the perceived grid, thus failing to provide the correct solution.

(3) **Algorithmic reasoning**: Language-based reasoners can struggle with complex multi-step reasoning tasks, such as inferring the connectivity (Figure 12) of a maze using the vertices and edges of the grid in pixel coordinates, or counting the number of objects located within a certain box.

## C  PRIMAL VISUAL DESCRIPTION (PVD) DETAILS

**PVD JSON schema definition:**    See Figure 13.

**Generation procedures (Single Object):**

- **Circle:** Randomly sample a center and a radius to draw a circle within the canvas.

- **Ellipse:** Randomly sample a center, a major axis, and a minor axis, then randomly rotate by an angle. Verify if the ellipse is largely within the canvas; if not, try again.

- **Rectangle:** Randomly sample a top-left corner, a width, and a height, then randomly rotate by an angle. Verify if the rectangle is largely within the canvas; if not, try again.

- **Triangle:** Randomly sample three points as vertices to draw a triangle. Check if the area is larger than a threshold; if not, try again.

- **Polygon:** Randomly sample $N \in [5, 20]$ points. Order the points with respect to the centroid so that no intersections will happen when connected with a polyline. Draw a polygon with the sampled points. Check if the polygon has an area larger than a threshold; if not, try again.

- **Path:** Randomly and iteratively sample $N \in [3, 16]$ points, connect the newly sampled point with the previous point to form a line segment. Verify if the newly added line segment does not intersect with any of the previous line segments; if yes, resample the point.

| Types | Schema | Example |
|-------|--------|---------|
| Circle | ```{     "type": "circle",     "center": [x, y],     "radius": r,     "color": [r, g, b],     "style": "filled shape" or "outlined shape",     "line_width": d (if style is "outlined") }``` | ```{     "type": "circle",     "center": [205, 210],     "radius": 117,     "color": [193, 190, 165],     "style": "outlined shape",     "line_width": 9 }``` |
| Ellipse | ```{     "type": "ellipse",     "center": [x, y],     "major_axis_length": l1,     "minor_axis_length": l2,     "rotation": o,     "color": [r, g, b],     "style": "filled shape" or "outlined shape",     "line_width": d (if style is "outlined") }``` | ```{     "type": "ellipse",     "center": [278, 166],     "major_axis_length": 147,     "minor_axis_length": 60,     "rotation": 16,     "color": [85, 220, 98],     "style": "filled shape" }``` |
| Rectangle Triangle Polygon | ```{     "type": "rectangle" or "triangle"         or "polygon",     "vertices": [[x1, y1], [x2, y2], ...]     "color": [r, g, b],     "style": "filled shape" or "outlined shape",     "line_width": d (if style is "outlined") }``` | ```{     "type": "triangle",     "vertices": [[452, 418], [298, 113],         [266, 255]],     "color": [165, 170, 141],     "style": "filled shape", }``` |
| Line Segment | ```{     "type": "line_segment",     "vertices": [[x1, y1], [x2, y2]]     "color": [r, g, b],     "line_width": d }``` | ```{     "type": "line_segment",     "vertices": [[822, 114], [93, 20]],     "color": [166, 32, 97],     "line_width": 10 }``` |
| Grid | ```{     "type": "grid",     "vertices": [[x1, y1], [x2, y2], ...],     "edges": [[[x1, y1], [x2, y2]], ...],     "color": [r, g, b],     "line_width": d }``` | ```{     "type": "grid",     "vertices": [[73, 214], [73, 640],         [215, 214], [215, 640]],     "edges": [[[73, 214], [73, 640]],         [[215, 214], [215, 640]],         [[73, 640], [215, 640]]]     "color": [23, 31, 120],     "line_width": 3 }``` |
| Path | ```{     "type": "path",     "vertices": [[x1, y1], [x2, y2], ...],     "edges": [[[x1, y1], [x2, y2]], ...],     "color": [r, g, b],     "line_width": d }``` | ```{     "type": "path",     "vertices": [[59, 69], [17, 330],         [61, 77]],     "edges": [[[59, 69], [17, 330]],         [[17, 330], [61, 77]]]     "color": [98, 28, 0],     "line_width": 5 }``` |
| Graph | ```{     "type": "line drawing",     "vertices": [[x1, y1], [x2, y2], ...],     "edges": [[[x1, y1], [x2, y2]], ...],     "color": [r, g, b],     "line_width": d }``` | ```{     "type": "line drawing",     "vertices": [[399, 497], [433, 823],         [483, 570], [531, 443], [534, 578]],     "edges": [[[399, 497], [483, 570]],         [[531, 443], [534, 578]],         [[483, 570], [534, 578]],         [[483, 570], [433, 823]],         [[534, 578], [433, 823]]]     "color": [254, 230, 139],     "line_width": 9 }``` |

Figure 13: PVD JSON schema definition.

- **Grid:** Sample a grid of points with a size $M \times N$ where $M, N \in [2, 6]$. First, use Depth First Search (DFS) algorithm to connect all grid vertices into a connected graph. Then randomly add additional edges between adjacent vertices.

- **Graph:** Randomly sample $N \in [4, 16]$ points. First, use Kruskal's algorithm (Kruskal, 1956) to find a Minimum Spanning Tree that connects all the points. Then randomly add additional edges to the graph.

**Generation procedures (Composition):** Iteratively draw shapes on the canvas chosen from the following set of object types: ["circle", "rectangle", "triangle", "line segment"]. After the first shape is drawn, at each iteration, the later shapes are constrained to have the same color as the previous shapes. We ensure overlap between the newly added shape and the previous shapes, while making

| | Style | Concept | # Instances |
|---|---|---|---|
| **Single Object** | Filled or Outlined | Circle | 10K |
| | | Ellipse | 10K |
| | | Rectangle | 10K |
| | | Triangle | 10K |
| | | Polygon | 20K |
| | | Line Segment | 10K |
| | | Grid | 10K |
| | | Path | 10K |
| | | Graph | 10K |
| **Composition** | Filled | Circle | 5K |
| | | Rectangle | 5K |
| | | Triangle | 5K |
| | | Line Segment | 5K |
| | Outlined | Circle | 10K |
| | | Rectangle | 10K |
| | | Triangle | 10K |
| | | Line Segment | 10K |
| | | **Total** | 160K |

Table 4: PVD 160K dataset statistics.

| | SSIM | DINOv2 Score | CLIP Score |
|---|---|---|---|
| w/o aug | 0.892 | 0.874 | 0.886 |
| w/ aug | **0.895** | **0.893** | **0.893** |

Table 5: Impact of the data augmentation (Gaussian Blur and Pixel Noise detailed in §2.2) on SVG-to-PVD model perception performance.

sure that the intersection ratio does not exceed a predefined threshold. This prevents cases where one shape entirely contains another, making it impossible to decode into individual Primal Visual Description elements.

**PVD 160K dataset:** Using the aforementioned generation procedure, we generate a large-scale dataset containing 160K ⟨SVG, PVD⟩ pairs for training the LLM-based SVG-to-PVD model. The detailed configuration can be found in Table 4.

**Data augmentation details:** To enhance the robustness of the SVG-to-PVD model to images with various sizes and quality, we introduce the following randomized data augmentation during data generation.

- **Random pixel noise**: Probability (how often to apply the augmentation): 0.1; Ratio range (what portion of the selected area will be filled with noise pixels): (0.01, 0.05); Intensity range (the intensity of the noise pixels): (0.1, 1.0); Dilate range (how many pixels will the selection area be extended from the boundary): (1, 3) in pixels; Noise size: (1, 3) in pixels.

- **Gaussian blur**: Probability (how often to apply the augmentation): 0.1; Radius: (0.1, 0.5).

Table 5 shows the ablation study with and without the data augmentations.

## D    FULL RESPONSE OF THE EXAMPLE IN FIGURE 2

See Figure 14 for the full input prompt and the generated response from GPT-4 on the 2×2 maze-solving task shown in Figure 2.

## E    TASK PROMPTS

Figure 15 shows the prompts for models with only image representations as visual inputs.

Figures 16-24 show the prompts for VDLM, where {perception} will be filled with the aggregated Primal Visual Description perception result, and the orange text are instance-specific inputs such as the question. For VDLM-mm, the original image input will be preserved and feed to the LMM reasoner along with the filled prompt. Since the reasoning in VDLM-txt is based solely on the PVD representation which is purely textual, task instructions that assume visual inputs can become ambiguous. For example, in the task Angle Classification, it is unclear which angle the question is referring to if we are only given the coordinates of two undirected edges. Therefore, we design task-specific prompts that remove such ambiguity. Another noteworthy point is that, in contrast to visual inputs that naturally accommodate a degree of imprecision, symbolic representations lack such inherent leniency. For instance, even if two line segments differ by only one pixel in length, they might be considered identical in visual representations, but symbolic representations would likely identify them as different. To reintroduce a level of tolerance in tasks that involve arithmetic reasoning, such as length comparison, we incorporate task-specific instructions to account for a reasonable margin of error, like 5%.

## F NEWLY CONSTRUCTED DOWNSTREAM TASK DATASETS

**Angle Classification.** We use the Geoclidean data generator[**] to generate images containing a single acute or obtuse angle with randomized orientations and ray lengths. The domain-specific language for generating the two concepts are shown as follows:

- Acute Angle:

```
"l1* = line(p1(), p2())",
"c1* = circle(p1(), p2())",
"c2* = circle(p2(), p1())",
"l2* = line(p3(c1, c2), p4(c1, c2))",
"l4 = line(p5(l1, l2), p7(l1))",
"l5 = line(p6(l2), p7(l1))"
```

- Obtuse Angle:

```
"l1* = line(p1(), p2())",
"c1* = circle(p1(), p2())",
"c2* = circle(p2(), p1())",
"l2* = line(p3(c1, c2), p4(c1, c2))",
"l3* = line(p5(l1, l2), p6(l2))",
"l4* = line(p5(l1, l2), p7(l1))",
"l5* = line(p6(l2), p7(l1))",
"l6* = line(p8(l3, l4), p9(l5))",
"l100* = line(p5(c1, c2), p10(l6))",
"c101* = circle(p5(c1, c2), p10(l6))",
"c102* = circle(p10(l6), p5(c1, c2))",
"l101* = line(p100(c101, c102), p101(c101,
c102))",
"l7 = line(p11(l100, l101), p6(l2))",
"l8 = line(p11(l100, 101), p7(l1))"
```

**Length Comparison.** We use matplotlib[††] to plot two non-intersecting line segments on a canvas. These line segments may either be of identical length or of differing lengths. In scenarios where the lengths vary, we ensure the discrepancy is substantial (exceeding 15% relative to the length of the shorter line segment) to ensure perceptibility. The orientation of each line segment is determined independently and randomly, being either horizontal or vertical.

**Maze Solving.** We leverage the maze-dataset package[‡‡] to generate 2D unsolved mazes along with their corresponding ground truth solutions. We use "circle" shape to denote the start position

---

[**]https://github.com/joyhsu0504/geoclidean_framework
[††]https://matplotlib.org/stable/
[‡‡]https://github.com/understanding-search/maze-dataset/tree/main

and "star" shape to denote the end position. We generate two subsets featuring 2×2 and 3×3 maze configurations.

# G   DATASET STATISTICS

|  |  | # Training Instances | # Eval Instances |
|---|---|---|---|
| **Probing Tasks** | Line or Angle | 10K | 1K |
|  | Angle Classification | 10K | 1000 |
|  | Length Comparison | 10K | 1000 |
|  | Clevr QA | 36K | 1000 |
|  | Shapeworld Scene | 15K | 100 |
|  | Maze Scene | 10K | 600 |
| **Zero-Shot Downstream Tasks** | Angle Classification | - | 100 |
|  | Length Comparison | - | 100 |
|  | Shapeworld Spatial Reasoning (2Obj) | - | 100 |
|  | Shapeworld Spatial Reasoning (MultiObj) | - | 100 |
|  | Shapeworld Superlative | - | 100 |
|  | NLVR | - | 100 |
|  | Geoclidean 2-shot Learning | - | 100 |
|  | 2×2 Maze Solving | - | 100 |
|  | 3×3 Maze Solving | - | 100 |
|  | VGBench-QA Category | - | 100 |
|  | VGBench-QA Color | - | 100 |
|  | VGBench-QA Usage | - | 100 |

Table 6: Statistics of the probing tasks (§ A.1) and the downstream tasks (§ 3). The GPT-4(V) zero-shot results on probing tasks are reported on 100 randomly sub-sampled instances from the entire eval split.

Detailed statistics of the probing tasks used in § A and the zero-shot downstream tasks mentioned in § 3 can be found in Table 6.

**Input Prompt**

The following JSON contains an approximated perception of a 2x2 maze. Each object (potentially including the background) is represented by a list of geometric shapes. If the object contains multiple shapes, it is a composite object. The (x, y) coordinates for the vertices and edges correspond to the width and height position in pixels, and (0, 0) is the top-left corner of the image.

--- perception ---
{'object_0': [{'type': 'rectangle', 'vertices': [[0.0, 336.0], [0.0, 0.0], [336.0, 0.0], [336.0, 336.0]], 'color': [253, 253, 253], 'style': 'filled shape'}], 'object_1': [{'type': 'grid', 'vertices': [[16, 16], [16, 164], [16, 312], [172, 16], [172, 164], [172, 312], [328, 16], [328, 164], [328, 312]], 'edges': [[[16, 16], [16, 164]], [[16, 16], [172, 16]], [[16, 164], [16, 312]], [[16, 164], [172, 164]], [[172, 16], [328, 16]], [[16, 312], [172, 312]], [[328, 16], [328, 164]], [[172, 312], [328, 312]], [[328, 164], [328, 312]]], 'color': [0, 0, 0], 'line_width': 10}], 'object_2': [{'type': 'circle', 'center': [94, 94], 'radius': 14, 'color': [255, 4, 4], 'style': 'filled shape'}], 'object_3': [{'type': 'triangle', 'vertices': [[262, 236], [222, 236], [253, 259]], 'color': [255, 7, 7], 'style': 'filled shape'}, {'type': 'triangle', 'vertices': [[242, 221], [231, 259], [250, 236]], 'color': [255, 7, 7], 'style': 'filled shape'}]}
------

In the 2x2 maze, walls are depicted by a grid of black lines between cells and along the perimeter. The maze cells are defined within the grid. The start position is marked by a red circle, and the end position by a red star. The position of each cell can be represented by coordinates (row, column), beginning from the top-left corner as (0, 0). Here, 'row' corresponds to the vertical dimension (height) of the image, and 'column' to the horizontal dimension (width).

Perform the following steps to solve the maze:
(1) Infer the connectivity of the cells using a connection list. For example, a 2x2 maze should have a `connection_list` containing two sublists with dimension 1x2 and 2x1. For i in range(0, 1) and j in range(0, 2), `connection_list[0][i][j]` is `True` if cell `(i, j)` is vertically connected to cell `(i+1, j)` without being seperated by a wall. Similarly, for j in range(0, 1) and i in range(0, 2), `connection_list[1][i][j]` is `True` if cell `(i, j)` is horizontally connected to cell `(i, j+1)` without being seperated by a wall.
(2) Infer the start position and end position of the maze in the row-column format.
(3) Solve the maze by finding a path from the start position to the end position.
You MUST provide your final answer, and the answer should follow this format: {"solution": "a list of (row, column) coordinates"}.

**GPT-4 Response**

To solve the maze based on the provided JSON data, let's break down the steps as required:

### Step 1: Infer the Connectivity of the Cells

Given the grid lines and their coordinates, we can infer the connectivity between cells. **The grid lines form the boundaries of a 2x2 maze, meaning there are 3 vertical lines and 3 horizontal lines that define the cells. By examining the edges in the JSON, we can determine where the walls are and thus infer which cells are connected.**

- Vertical connections (`connection_list[0]`): There are no vertical walls inside the maze, so both cells in the first column are connected to their respective cells in the second column.
- Horizontal connections (`connection_list[1]`): There is a horizontal wall between the two cells in the first row, meaning there's no direct connection between them. However, the cells in the second row are connected.

Therefore, the `connection_list` can be represented as:
- **Vertical connections: `[[True, True]]`**
- **Horizontal connections: `[[False], [True]]`**

### Step 2: Infer the Start and End Positions

- The **start position** is marked by a **red circle**, which is located at approximately `[94, 94]`. This places it in the **top-left cell, or `(0, 0)`**.
- The **end position** is marked by **a composite red shape resembling a star, located in the bottom-right cell**, which can be inferred from the coordinates of the triangles forming the star shape. This **places the end position at `(1, 1)`**.

### Step 3: Solve the Maze

Given the connectivity and the start/end positions, we can find a path from the start to the end. The path, considering the connectivity and the positions, would be:
1. Start at `(0, 0)`.
2. Move right to `(0, 1)`.
3. Move down to `(1, 1)`.

Thus, the solution path is: `[(0, 0), (0, 1), (1, 1)]`.

### Final Answer

```json
{"solution": "[(0, 0), (0, 1), (1, 1)]"}
```

Figure 14: Full input prompt and GPT-4 response of the 2x2 maze solving example in Figure 2. The blue part in the input prompt indicates the generated Primal Visual Description (PVD) of the entire image.

Is this an acute angle or an obtuse angle? Answer with "Acute" or "Obtuse".

Acute-or-Obtuse

Are these two lines of equal length? Choose "Yes" or "No".

Length-Comparison

Given a 2d scene containing objects with the following attributes:
--- Attribute Ontology ---
- color: ['red', 'green', 'blue', 'yellow', 'magenta', 'cyan', 'gray']
- shape: ['square', 'rectangle', 'triangle', 'pentagon', 'cross', 'circle', 'semicircle', 'ellipse']
------

Think step by step and answer the following question:
{question}

You MUST provide your final answer, and the answer should follow this format: {{"answer": "yes" or "no"}}

Shapeworld

Given an image containing three boxes with light grey background, horizontally layed out. The boxes are seperated by two dark grey rectangles placed vertically, which are referred to as walls. An "edge" is referred to as the boundary of the image. A "base edge" is referred to as the bottom boundary. If an object's boundary is located very close to an edge or a wall (e.g., within 10 pixels), it is considered as "touching". Each box contains a set of shapes. There are two types of images: "Tower", and "Scatter". In "Tower" images, each box contains only squares stacked in towers with up to 4 squares. In "Scatter" images, each box contains scattered objects of different sizes and shapes.

Think step by step to answer the following question:
{question}

You MUST provide your final answer, and the answer should follow this format: {{"answer": "yes" or "no"}}

NLVR

The top part of the scene provides {n_shot} reference examples of a Euclidean geometry concept. A Euclidean geometry concept consists of a composition of one or more primitive geometric shapes, such as circles and line segments, with some constraints. These constraints include but are not limited to lengths, angles, and spatial relationships between the primitive shapes, for example, two perpendicular line segments or an equilateral triangle, etc.
The bottom part of the scene presents a test example, separated from the top part by a red horizontal line.
First, identify the pattern and constraints of the reference and test concepts.
Then, determine if the test example depicts the same concept as the reference examples.

You MUST provide your final answer, and the answer should follow this format: {{"answer": "yes" or "no"}}.

Geoclidean-2Shot

Given a {n}x{n} maze, where the walls are depicted by a grid of black lines between cells and along the perimeter. The maze cells are defined within the grid. The start position is marked by a red circle, and the end position by a red star. The position of each cell can be represented by coordinates (row, column), beginning from the top-left corner as (0, 0). Here, 'row' corresponds to the vertical dimension (height) of the image, and 'column' to the horizontal dimension (width).

Solve the maze by finding a path from the start position to the end position.
You MUST provide your final answer, and the answer should follow this format: {{"solution": "a list of (row, column) coordinates"}}.

Maze-Solve

Given an image containing a SVG graphic, think step by step and answer the following question:
{question}

{options}

You MUST provide your final answer, and the answer should follow this format: {{"answer": choose from "A", "B", "C", "D"}}

VGBench QA

Figure 15: Prompts for zero-shot downstream tasks with image input

The following JSON contains an approximated perception of a 2d scene. Each object (potentially including the background) is represented by a list of geometric shapes. If the object contain multiple shapes, it is a composite object. The (x, y) coordinates are in pixels, and (0, 0) is the top-left corner of the image.

--- perception ---
{perception}
------

Answer the following question:
Is this an acute angle or an obtuse angle?

Note that if the perception result includes a path, the angle is defined with the midpoint as the vertex, and the rays extend from the midpoint to both the head and end points. For example, if a path from A to B to C is perceived, the angle is defined as the angle between the vector BA and BC.
You MUST provide your final answer, and the answer should follow this format: {{"answer": "acute" or "obtuse"}}.

Figure 16: Prompt for task Angle Classification with Primal Visual Description perception input.

The following JSON contains an approximated perception of a 2d scene. Each object (potentially including the background) is represented by a list of geometric shapes. If the object contain multiple shapes, it is a composite object. The (x, y) coordinates are in pixels, and (0, 0) is the top-left corner of the image.

--- perception ---
{perception}
------

Answer the following question:
Are these two lines of equal length?

Note that perception can be noisy. A 5% offset in the measurements is acceptable. You MUST provide your final answer, and the answer should follow this format: {{"answer": "yes" or "no"}}

Figure 17: Prompt for task Length Comparison with Primal Visual Description perception input.

Given a 2d scene containing objects with the following attributes:
--- Attribute Ontology ---
- color: ['red', 'green', 'blue', 'yellow', 'magenta', 'cyan', 'gray']
- shape: ['square', 'rectangle', 'triangle', 'pentagon', 'cross', 'circle', 'semicircle', 'ellipse']
------

The following JSON contains an approximated perception of the scene. Each object (potentially including the background) is represented by a list of geometric shapes. If the object contain multiple shapes, it is a composite object. The (x, y) coordinates are in pixels, and (0, 0) is the top-left corner of the image.

--- perception ---
{perception}
------

Note that the perception can be noisy. First identify the best matching shape type and the color type from the ontology for each perceived object. For composite objects, please match the entire composition to one of the most probable objects in the ontology. Make educated guesses if necessary. Then, think step by step and answer the following question:
{question}

You MUST provide your final answer, and the answer should follow this format: {{"answer": "yes" or "no"}}

Figure 18: Prompt for task Shapeworld Spatial Reasoning (2Obj) with Primal Visual Description perception input.

Given a 2d scene containing objects with the following attributes:
--- Attribute Ontology ---
- color: ['red', 'green', 'blue', 'yellow', 'magenta', 'cyan', 'gray']
- shape: ['square', 'rectangle', 'triangle', 'pentagon', 'cross', 'circle', 'semicircle', 'ellipse']
------

The following JSON contains an approximated perception of the scene. Each object (potentially including the background) is represented by a list of geometric shapes. If the object contain multiple shapes, it is a composite object. The (x, y) coordinates are in pixels, and (0, 0) is the top-left corner of the image. If two objects overlap, the one with the larger index is considered to be in front of the other.

--- perception ---
{perception}
------

Note that the perception can be noisy. First identify the best matching shape type and the color type from the ontology for each perceived object. For composite objects, please match the entire composition to one of the most probable objects in the ontology. Make educated guesses if necessary. Then, think step by step and answer the following question:
{question}

You MUST provide your final answer, and the answer should follow this format: {{"answer": "yes" or "no"}}

Figure 19: Prompt for task Shapeworld Spatial Reasoning (MultiObj) with Primal Visual Description perception input.

Given a 2d scene containing objects with the following attributes:
--- Attribute Ontology ---
- color: ['red', 'green', 'blue', 'yellow', 'magenta', 'cyan', 'gray']
- shape: ['square', 'rectangle', 'triangle', 'pentagon', 'cross', 'circle', 'semicircle', 'ellipse']
------

The following JSON contains an approximated perception of the scene. Each object (potentially including the background) is represented by a list of geometric shapes. If the object contain multiple shapes, it is a composite object. The (x, y) coordinates are in pixels, and (0, 0) is the top-left corner of the image. The lowermost object has the largest y-coordinate, and the rightmost object has the largest x-coordinate.

--- perception ---
{perception}
------

Note that the perception can be noisy. First identify the best matching shape type and the color type from the ontology for each perceived object. For composite objects, please match the entire composition to one of the most probable objects in the ontology. Make educated guesses if necessary. Then, think step by step and answer the following question:
{question}

You MUST provide your final answer, and the answer should follow this format: {{"answer": "yes" or "no"}}

Figure 20: Prompt for task Shapeworld Superlative with Primal Visual Description perception input.

Given an image containing three boxes with light grey background, horizontally laid out. The boxes are separated by two dark grey rectangles placed vertically, which are referred to as walls. An "edge" is referred to as the boundary of the image. A "base edge" is referred to as the bottom boundary. If an object's boundary is located very close to an edge or a wall (e.g., within 10 pixels), it is considered as "touching". Each box contains a set of shapes. There are two types of images: "Tower", and "Scatter". In "Tower" images, each box contains only squares stacked in towers with up to 4 squares. In "Scatter" images, each box contains scattered objects of different sizes and shapes.

The following JSON contains an approximated perception of the image. Each object (potentially including the background) is represented by a list of geometric shapes. If the object contain multiple shapes, it is a composite object. The (x, y) coordinates are in pixels, and (0, 0) is the top-left corner of the image.

--- perception ---
{perception}
------

Now, identify the content in each box based on the perception result, and then think step by step to answer the following question:
{question}

You MUST provide your final answer, and the answer should follow this format: {{"answer": "yes" or "no"}}

Figure 21: Prompt for task NLVR with Primal Visual Description perception input.

The following JSON contains an approximated perception of the image. Each object (potentially including the background) is represented by a list of geometric shapes. If the object contain multiple shapes, it is a composite object. The (x, y) coordinates are in pixels, and (0, 0) is the top-left corner of the image.

--- perception ---
{perception}
------

The top part of the scene provides {n_shot} reference examples of a Euclidean geometry concept. A Euclidean geometry concept consists of a composition of one or more primitive geometric shapes, such as circles and line segments, with some constraints. These constraints include but are not limited to lengths, angles, and spatial relationships between the primitive shapes, for example, two perpendicular line segments or an equilateral triangle, etc.
The bottom part of the scene presents a test example, separated from the top part by a red horizontal line.
First, identify the pattern and constraints of the reference and test concepts based on the perception result. Note that the perception can be noisy. Make educated guesses if necessary.
Then, determine if the test example depicts the same concept as the reference examples.

You MUST provide your final answer, and the answer should follow this format: {{"answer": "yes" or "no"}}.

Figure 22: Prompt for task Geoclidean 2-shot Learning with Primal Visual Description perception input.

The following JSON contains an approximated perception of a {n}x{n} maze. Each object (potentially including the background) is represented by a list of geometric shapes. If the object contains multiple shapes, it is a composite object. The (x, y) coordinates for the vertices and edges correspond to the width and height position in pixels, and (0, 0) is the top-left corner of the image.

--- perception ---
{perception}
------

In the {n}x{n} maze, walls are depicted by a grid of black lines between cells and along the perimeter. The maze cells are defined within the grid. The start position is marked by a red circle, and the end position by a red star. The position of each cell can be represented by coordinates (row, column), beginning from the top-left corner as (0, 0). Here, 'row' corresponds to the vertical dimension (height) of the image, and 'column' to the horizontal dimension (width).

Perform the following steps to solve the maze:
(1) Infer the connectivity of the cells using a connection list. For example, a {n}x{n} maze should have a `connection_list` containing two sublisits with dimension {m}x{n} and {n}x{m}. For i in range(0, {m}) and j in range(0, {n}), `connection_list[0][i][j]` is `True` if cell `(i, j)` is vertically connected to cell `(i+1, j)` without being seperated by a wall. Similarly, for j in range(0, {m}) and i in range(0, {n}), `connection_list[1][i][j]` is `True` if cell `(i, j)` is horizontally connected to cell `(i, j+1)` without being seperated by a wall.
(2) Infer the start position and end position of the maze in the row-column format.
(3) Solve the maze by finding a path from the start position to the end position.
You MUST provide your final answer, and the answer should follow this format: {{"solution": "a list of (row, column) coordinates"}}.

Figure 23: Prompt for task Maze Solving with Primal Visual Description perception input.

Given an image containing a SVG graphic, think step by step and answer the following question:
{question}

{options}

------
The following JSON contains an approximated reference perception of the image. Each object (potentially including the background) is represented by a list of geometric shapes. If the object contain multiple shapes, it is a composite object. The (x, y) coordinates are in pixels, and (0, 0) is the top-left corner of the image.

--- reference perception ---
{perception}
------

Note that the reference perception can be noisy. Refer to the reference perception when necessary for answering the question.

You MUST provide your final answer, and the answer should follow this format: {{"answer": choose from "A", "B", "C", "D"}}

Figure 24: Prompt for VGBench-QA tasks with Primal Visual Description perception input.

