# OpenReview forum: "Visually Descriptive Language Model for Vector Graphics Reasoning"
_ICLR.cc/2025/Conference — ICLR 2025 Conference Withdrawn Submission_

### Official Review · Reviewer_hw2c · 2024-10-29

**Soundness:** 3
**Presentation:** 3
**Contribution:** 2
**Rating:** 5
**Confidence:** 4

**Summary:**

This paper proposes a method called VDLM, designed to improve large vision-language models capabilities to understand and reason about fine-grained visual concepts, specifically those well-suited to vector graphic representations. VDLM first converts an input image into a SVG format using a rule-based encoder. This SVG is then preprocessed and translated by a fine-tuned language model (e.g., Mistral-7b) into a structured JSON-like format called "Primal Visual Description" (PVD), which serves as an intermediate representation. Finally, this PVD is input into large vision-language models (e.g., GPT-4). The authors demonstrate that this pipeline improves performance of the base model (e.g. GPT4o) vs using a raw image or a raw/unprocessed SVG encoding.

**Strengths:**

* The paper focuses on an area of significant impact: improving visual reasoning/understanding in large vision-language models.

* The writing is clear and well-structured, making the methodology easy to follow.

* The proposed method achieves notable improvements in several evaluated tasks.

**Weaknesses:**

In my view, the main weakness of the paper is that, while it aims to "enable precise  visual perception and facilitate high level reasoning", the approach does not actually relies on visual perception. Instead, it sidesteps the need for genuine visual comprehension by essentially translating an image into text and performing all reasoning in text, limiting effectiveness for complex imagery requiring true visual inspection. Figure 5 supports this: only the GPT models show improvement, suggesting limited applicability unless the model’s text-based understanding is SoTA.

SVG Limitations:  While the approach may work on some SVG, the claim in the abstract that SVG can accurately capture detailed visual scenes seems overstated, particularly within the context of this study.

Questionable Fine-Tuning Claim: The authors claim in line 085 “finetuning a model to reason about raw SVG codes can be inefficient and infeasible without corresponding task-specific annotations.”  This claim appears unsubstantiated, as fine-tuning an existing large multimodal model (e.g., LLaMA 3.2 11B, LLaVA) on the synthetic data generated for the PVD translator could potentially improve visual perception on such imagery without bypassing the need for vision.

Key questions and suggestions:

1. Can the SVG-PVD translator be replaced by a model like GPT-4o using few-shot in-context learning on synthetic data, avoiding the need for extensive fine-tuning? This could serve as a viable baseline.

2. Given the authors opt for fine-tuning, a comparison where a model similar to Mistral-7b is fine-tuned to solve the task end-to-end rather than translating into the PVD language should be shown.

Additionally, in Table 1, presenting GPT-4o’s performance with text only would help isolate the effect of model power versus the impact of image integration, rather than conflating both under a single comparison (e.g. VDLM-txt GPT4 vs. VDLM-txt GPT4o). Based on recent findings in the literature, I might guess that the image is adding very little given the current content and most of the information needed available in text.

**Questions:**

see above

---

> ### Author Response · Authors · 2024-12-02
>
> Thank you for the feedback and insightful reviews. We are glad that you find the focus to be impactful and that the method has notable improvements.
>
> **W1: Justification on the key idea**
>
> The key idea of this paper is to bridge the gap between low-level perception and high-level reasoning. We argue that visual perception is not solely defined by how visual information is represented. We demonstrate that the proposed PVD representation complements popular visual representations, such as CLIP, which are semantic-centric and less effective in capturing low-level details.
>
> We believe leveraging the strong text-based reasoning capabilities of existing foundation models is not a limitation. Humans also rely on abstract language and symbols to understand the visual world. Moreover, the fact that VDLM benefits only emerge in strong LMMs is analogous to human development, where abstract reasoning typically begins after the age of three.
>
> **W2: SVG Limitations**
>
> As mentioned at the beginning of the abstract, we focus on vector graphics, a particularly challenging visual domain for low-level visual reasoning in LMMs. As discussed in the future work section, we acknowledge that while SVGs faithfully capture 2D visual scenes, they are limited in representing 3D scenes and natural images. We leave these extensions for future work.
>
> **W3: Questionable Fine-Tuning Claim**
>
> The claim is that directly fine-tuning a model on “raw SVG codes” to “answers” is infeasible without a large amount of task-specific “SVG-Question-Answer” pairs and inefficient even with sufficient data. The reviewer suggests training a VLM on SVG-PVD synthesized data, which could work as an alternative SVG-PVD generator. However, such an approach still lacks the ability to generalize across different tasks since the synthesized data is task-agnostic.
>
> **Q1: Can GPT-4o already do the SVG-to-PVD translation**
>
> We attempted this in our preliminary experiments. As of the time of writing, GPT-4o cannot interpret raw SVG codes, even with few-shot examples. We will add these results in the revised version.
>
> **Q2: Fine-tuning Mistral-7b end-to-end**
>
> A key motivation for the modularized VDLM design—separating PVD perception and LMM reasoning—is the lack of task-specific, end-to-end data from SVG to questions and answers. For most tasks in our benchmark, such end-to-end data is unavailable. Moreover, such end-to-end training constrains the model’s ability to generalize to unseen tasks. Consequently, we propose a disentangled perception-and-reasoning framework.
>
> **Q3: Suggestion on adding VDLM-txt w/ GPT-4o text-only**
>
> Thank you for the suggestion. We agree that adding this would make the results more comprehensive. We will include this in the next revision. Our observation is that the text-based capabilities of GPT-4o are comparable to those of GPT-4. Thus, one can expect similar performance for VDLM-txt with GPT-4.

---

### Official Review · Reviewer_DbWW · 2024-10-30

**Soundness:** 3
**Presentation:** 3
**Contribution:** 3
**Rating:** 5
**Confidence:** 4

**Summary:**

This work focuses on vector graphics and proposes primal visual description (PVD) to better perceive and reasoning about 2D objects and shapes. Specifically, they propose PVD technique translates SVGs into a text-bassed abstraction aomprising primitive attributes along with their corresponding values. In this way, the PVD can replace the visual embeddings and directly participate in the LLM's reasoning process, improving the interpretability due to the disentangled perception and reasnoing processes.

**Strengths:**

1. The writing in this work is excellent, with clear logic that makes it easy to follow.

2. The experimental and analysis sections in this work are thorough and well-developed.

3. The concept of PVD is intuitive, simple, and effective.

**Weaknesses:**

1. The scope of application domains and evaluation data considered by the authors is too narrow. Although the authors mention web and OS environments as application scenarios in the abstract, the experimental evaluation is limited to simple SVG images. This may restrict the impact and significance of the work. In real-world applications, we cannot always assume that SVG-format images will be available.

2. The evaluation of high-level visual reasoning tasks is insufficient. First, only one benchmark was included in the validation. Secondly, the study only examined improvements for GPT-4V with VDLM support, which raises questions about whether other open-source, less capable LMMs (e.g., LLaVA-1.5, Qwen-VL) could benefit from VDLM as well.

**Questions:**

1. What does the "VF" mean in Figure 1(b)? Does it denote the visual features?

2. Lacking discussion about related work [A]. It also highlights the disentangling of perception and reasoning for LMMs.

[A] Prism: A Framework for Decoupling and Assessing the Capabilities of VLMs. NeurIPS 2024

I will consider raising the score if my concerns can be well addressed.

---

> ### Author Response · Authors · 2024-12-02
>
> Thank you for the feedback and insightful reviews. We are glad that you find our writing to be excellent and our experiments to be thorough.
>
> **W1: Scope of this work**
>
> We would like to clarify that all instances of “vector graphics” input in this paper refer to raster images. We do not make any assumptions about the input image format, i.e., we do not assume access to the underlying vector codes, such as the ground truth SVG.
>
> As the first attempt to build a descriptive intermediate representation to address the long-standing low-level visual reasoning problem, we focus on demonstrating proof of concept. We conduct controlled experiments on standalone vector graphics to reveal the issue and show the potential of VDLM. Nevertheless, we acknowledge that the PVD ontology is not yet perfect and encourage future work. There are several ways to make the approach more general, including (a) incorporating human-created vector graphics with procedural annotations to diversify the ontology and training dataset; (b) involving large language models in the data synthesis phase to enrich diversity; and (c) incorporating visual search during inference to convert focused regions/parts into PVD and perform multi-step reasoning.
>
> A simple pipeline to apply VDLM to web and OS environments is: (a) using icon/object grounding models and OCR models (such as those in [1]) to detect icons and text, and then (b) feeding each icon/object into VDLM for PVD description and visual reasoning.
>
>
> **W2: Evaluation on high-level visual reasoning tasks**
>
> The focus of this work is on low-level visual reasoning. However, we include high-level reasoning tasks to demonstrate that VDLM can preserve high-level reasoning abilities while improving low-level reasoning.
>
> As shown in Figure 5, we applied VDLM to open-source models, such as Llava-v1.5. We observe that the benefits only emerge in LMMs with sufficiently strong text-reasoning abilities. This is expected and is analogous to human development, where the emergence of abstract reasoning typically begins after three years of age.
>
> **Q1: “VF” meaning**
>
> Thanks for pointing out the confusion! Yes, it denotes “Visual Features” as in Figure 1(a). We will make this clearer in the caption.
>
> **Q2: Related work**
>
> Thanks for the pointer to the related work. We will include a discussion in the next revision.
>
> References
> 1. Mobile-Agent-v2: Mobile Device Operation Assistant with Effective Navigation via Multi-Agent Collaboration

---

### Official Review · Reviewer_673U · 2024-11-01

**Soundness:** 2
**Presentation:** 2
**Contribution:** 3
**Rating:** 3
**Confidence:** 4

**Summary:**

This paper introduces a novel approach to enhance fine-grained image understanding by transforming raster images into a new vector graphics format called PVD, which consists of 9 distinct primitives. The authors propose a two-stage process: first, converting the raster image to an SVG format using a rule-based model, and second, converting the SVG to the proposed PVD format using a self-trained LLM. By incorporating the PVD representation into the LLM’s context, the authors report improved performance in fine-grained image understanding tasks. The paper also provides a comparative analysis with baseline approaches that input either raw images into VLMs or SVG representations into LLMs, highlighting the advantages of their method.

**Strengths:**

1. This work appears to be the first attempt to enhance foundation models' fine-grained image understanding capabilities by converting images into vector-based formats.
2. The introduction of the PVD vector graphics format offers a structure that seems better aligned with large language models.
3. Significant performance gains are observed when using the proposed PVD-based approach compared to directly using VLMs.

**Weaknesses:**

1. The rationale behind the two-stage transformation process (from PNG to SVG via a rule-based algorithm, and then from SVG to PVD via a self-trained LLM) requires further clarification. Specifically: (a) What fundamental distinctions, other than syntax, does PVD offer compared to SVG? (b) Why is an LLM necessary for the second step, given that SVG is a structured, well-defined format that could potentially be converted to PVD with rule-based transformations? (c) Why not directly modify the rule-based raster-to-vector algorithm to output PVD instead of SVG?
2. The proposed PVD format’s definition could benefit from further clarity, as there appears to be overlap among the 9 primitives (e.g., "composition-filled" could overlap with "polygon"; "path" and "grid" might overlap with "line segment"). Introducing a hierarchical structure might provide a clearer and more effective organization.
3. The approach relies on synthesized data for training the LLM to convert SVG to PVD, which might limit the diversity and generalizability of the model’s performance.
4. Some terms in the paper could be refined for accuracy. For instance, the term "vector graphics" should refer to images defined by geometric primitives, such as SVG and PVD. In contrast, this work use that term "vector graphics" to refer to images rendered by SVG (which is the raster image in fact) (e.g., in the title and L91). Additionally, conflating "vector graphics" and "SVG" may be misleading (e.g., in L21–L22), as SVG is just one of several vector graphics formats.

**Questions:**

1. The results in Table 3 (Appendix) seem inconsistent with those in Table 1. If I understand correctly, Table 1 reports that GPT-4V achieved 58% in Angle Classification (AC) and 64% in Length Comparison (LC) with image inputs. However, Table 3 lists GPT-4V as achieving 58% in Angle Classification (AC) and only 57% in Length Comparison (LC). Could you clarify this discrepancy?
2. See questions raised in the weakness.

---

> ### Author Response · Authors · 2024-12-02
>
> Thank you for the feedback and insightful reviews. We are glad that you find our approach to be pioneering and the improvements to be significant.
>
> **W1: Rationale behind the two-stage process**
>
> The key idea of introducing the two-stage design (SVG → PVD → Answer) is to bridge the gap between low-level detailed perception and high-level reasoning. SVG can faithfully capture low-level details but is NOT directly understandable by LMMs/LMMs.
>
> (a) The automatically encoded SVGs from raster images are very verbose and noisy, making them uninterpretable even by the strongest LLMs/LMMs. For example, a simple rectangle in raw SVG format can require tens to hundreds of points, while in PVD, it only requires four points. Essentially, PVD is a cleaner, higher-level abstraction that facilitates reasoning while retaining the most important low-level features, such as the coordinates of vertices.
>
> (b, c) As mentioned earlier, although the encoded raw SVGs are structured, they are highly noisy. It is impractical to use a rule-based algorithm to convert SVGs into PVD. A language model is necessary to handle the complex variances involved. Similarly, directly modifying the rule-based VTracer to generate PVD is infeasible, as it lacks awareness of the semantics of shapes.
>
> **W2: Justification of the Ontology**
>
> 1. We construct the ontology primarily based on the primitives humans use to perceive vector graphics. While a path can technically be represented as a sequence of lines, this approach neglects the implicit constraint that the start and end vertices are connected. Humans naturally include such constraints during perception. This also allows the model to handle complex concepts more effectively.
>
> 2. We did not enforce strict non-overlapping constraints in the ontology because it is intuitive that a final concept may have multiple valid compositions from primitives. For instance, a “cross” can be interpreted either as a “polygon” or as a “composition” of two rectangles. We allow for these variances when designing the ontology.
>
> **W3: Generalizability**
>
> As the first attempt to build a descriptive intermediate representation for addressing the long-standing low-level visual reasoning problem, we focus on demonstrating proof of concept. As discussed in the Results section, we are aware that the current PVD representation is not expressive enough and call for future work. There are several ways we are considering to make the approach more general, including (1) incorporating human-created vector graphics with procedural annotations to diversify the ontology and training dataset; (2) involving large language models in the data synthesis phase to enrich diversity; and (3) incorporating visual search during inference to convert focused regions/parts into PVD and perform multi-step reasoning.
>
> **W4: Terms**
>
> Thanks for pointing out this potential confusion! Yes, the “vector graphics” in this paper refers to the rasterized images without assuming access to the underlying vector codes. We will make this clear in the next revision.
>
> **Q1: Scores in Appendix**
>
> Thanks for the catch! That is indeed a typo, we will fix it in the next revision.

---

### Official Review · Reviewer_8s4N · 2024-11-07

**Soundness:** 3
**Presentation:** 3
**Contribution:** 2
**Rating:** 5
**Confidence:** 2

**Summary:**

The paper introduces VDLM that aims to decrease inaccuracies associated with modern VLMs by USING an intermediate representation Primal Visual Description (PVD), primitive geometry object descriptor that can help ground the interpretation of the image and increase the usability of the model on perception and reasoning tasks downstream tasks involving vector graphics. More specifically, the paper identifies Low-level visual reasoning tasks and High-level visual reasoning tasks, introducing benchmarks for evaluating the former and using VGBench-QA for the latter, and showing performance gains for both text and multimodal versions of VDLM.

**Strengths:**

- Interesting issue: The VLMs are known to struggle with low-level details in image analysis, making a low-resource solution like VDLM impactful.
- Benchmarking dataset: To the best of my knowledge the dataset is unique and is a significant contribution that could serve as a good benchmark for quantitative evaluation of current and future VLMs.
- PVD approach: Is intuitive and encodes a lot of information about the object in a relatively compact manner.

**Weaknesses:**

- Practical usability: The Ontology of PVD is limited and seems like it would be hard to expand to more general use cases, furthermore it seems like PVD use would strain the context length significantly in any more complex use cases, thus while PVDs might be learnable in task-agnostic setting I am concerned about the generalizability of VDLM.
- Limited baselines: The paper reports numbers for Variations of GPT, LLaMA, ViperGPT models, however there are no comparisons with any symbolic approaches, making the positioning of the results hard to place.

**Questions:**

- How would the method extend to SVG complexity found in websites and most usual use cases?
- How well could the method handle occlusion between distinct shapes?
- Could the authors provide further reasoning behind their choice of baselines? Why were symbolic reasoning approaches not included in the comparison?

---

> ### Author Response · Authors · 2024-12-02
>
> Thank you for the feedback and insightful reviews. We are glad that you find our identified issue to be interesting and our proposed method to be impactful. Below, we address your suggestions and questions.
>
> **W1: Practical usability**
>
> As the first attempt to build a descriptive intermediate representation for addressing the long-standing low-level visual reasoning problem, we focus on demonstrating proof of concept. As discussed in the Results section, we are aware that the PVD ontology is not yet perfect and call for future work. There are several ways we are considering to make PVD more general, including (1) incorporating human-created vector graphics with procedural annotations to diversify the ontology and training dataset; (2) involving large language models in the data synthesis phase to enrich diversity; and (3) incorporating visual search during inference to convert focused regions/parts into PVD and perform multi-step reasoning.
>
> The context length issue is valid but will likely be less of a concern in the future, as the latest LMMs offer extended contexts reaching up to 200K tokens (e.g., Claude 3.5). Additionally, we can potentially decompose complex problems into multiple perception-reasoning steps during inference.
>
> We will add these discussions to the next revision.
>
> **W2: Baselines**
>
> The focus of this work is to evaluate and improve LMMs’ low-level visual reasoning abilities, so we primarily consider LMM-based baselines. As mentioned in the Related Work section, most neuro-symbolic prior work [2, 3] is confined to dataset-specific programs and requires supervised training, making them unsuitable for direct comparison.
>
> **Q1: How to extend to SVG complexity found in websites and most usual use cases?**
>
> A simple pipeline would involve (1) using icon/object grounding models and OCR models (such as those in [1]) to detect icons and text, and then (2) feeding each icon/object into VDLM for PVD description and visual reasoning.
>
> **Q2: How well could the method handle occlusion between distinct shapes?**
>
> Thank you for raising this interesting challenge. Currently, if a Shape A is occluded by another Shape B (with a different color), the VTracer would generate two SVG paths for Shape A, and the VDLM framework would include two separate PVD objects for Shape A. Since VDLM-mm includes the original pixel inputs, the LMM reasoner can reason over the two PVDs as parts of Shape A. However, ideally, the PVD representation should reflect a more coherent object.
>
> Potential ways to address this challenge include (1) incorporating occlusion scenarios into the SVG-to-PVD training data and (2) leveraging additional segmentation models during inference to aid in decomposing SVG paths for more coherent object representations.
>
> **Q3: Justification on baselines**
>
> See W2
>
> References
> 1. Mobile-Agent-v2: Mobile Device Operation Assistant with Effective Navigation via Multi-Agent Collaboration
> 2. Neural-Symbolic VQA: Disentangling Reasoning from Vision and Language Understanding
> 3. Neural Scene De-rendering

---

### Note · Authors · 2024-11-13

I have read and agree with the venue's withdrawal policy on behalf of myself and my co-authors.

---

> ### Note · Program_Chairs · 2024-12-02
>
> **Comment:**
>
> Withdrawal reversed temporarily.
>
> **Revert Withdrawal Confirmation:**
>
> We approve the reversion of withdrawn submission.

---

### Note · Authors · 2024-12-02

**Comment:**

Requested by authors.

**Withdrawal Confirmation:**

I have read and agree with the venue's withdrawal policy on behalf of myself and my co-authors.